# Cross-Chirality Generalization by Axial Vectors for Hetero-Chiral Protein-Peptide Interaction Design

Ziyi Yang [* 1 2]   Zitong Tian [* 3]   Yinjun Jia [* 4]   Tianyi Zhang [5 6]   Jiqing Zheng [1]   Hao Wang [1]   Yubu Su [1]   Juncai He [3 7]   Lei Liu [1 8 9 10]   Yanyan Lan [4 11]

## Abstract

D-peptide binders targeting L-proteins have promising therapeutic potential. Despite rapid advances in machine learning-based target-conditioned peptide design, generating D-peptide binders remains largely unexplored. In this work, we show that by injecting axial features to $E(3)$-equivariant (polar) vector features, it is feasible to achieve cross-chirality generalization from homo-chiral (L–L) training data to hetero-chiral (D–L) design tasks. By implementing this method within a latent diffusion model, we achieved D-peptide binder design that not only outperforms existing tools in *in silico* benchmarks, but also demonstrates efficacy in wet-lab validation. To our knowledge, our approach represents the first wet-lab validated generative AI for the *de novo* design of D-peptide binders, offering new perspectives on handling chirality in protein design. Codes are available at https://github.com/YZY010418/PepMirror.

---

[*]Equal contribution [1]Department of Chemistry, Tsinghua University, Beijing, China [2]Anew Labs, Shanghai, China [3]Qiuzhen College, Tsinghua University, Beijing, China [4]Institute for AI Industry Research (AIR), Tsinghua University, Beijing, China [5]School of Life Sciences, Tsinghua University, Beijing, China [6]AI Industry Research Innovation Center,Wuxi Research Institute for Applied Technologies, Tsinghua University [7]Yau Mathematical Sciences Center, Tsinghua University, Beijing, China [8]Tsinghua-Peking Center for Life Sciences, Beijing, China [9]Ministry of Education Key Laboratory of Bioorganic Phosphorus Chemistry and Chemical Biology, Tsinghua University, Beijing, China [10]Center for Synthetic and Systems Biology, Tsinghua University, Beijing, China [11]Beijing Frontier Research Center for Biological Structure, Tsinghua University, Beijing, China. Correspondence to: Juncai He <jche@tsinghua.edu.cn>, Lei Liu <lliu@mail.tsinghua.edu.cn>, Yanyan Lan <lanyanyan@air.tsinghua.edu.cn>.

*Proceedings of the $43^{rd}$ International Conference on Machine Learning*, Seoul, South Korea. PMLR 306, 2026. Copyright 2026 by the author(s).

## 1. Introduction

A hallmark of proteins is their homo-chirality: nature predominantly uses L-amino acids to construct proteins within living organisms (Pasteur, 1848; Blackmond, 2010). In this context, D-peptides become advantaged molecules for therapeutics due to their intrinsic orthogonality in an all-L system. D-peptides cannot be recognized as substrates by proteases, thereby exhibiting prolonged *in vivo* half-life (Kremsmayr et al., 2022). Besides, the bioorthogonality of D-peptides reduces their immunogenecity and risks of drug-drug interactions (DDI) (Lander et al., 2023; Qi et al., 2024). Conventionally, D-peptide binders are identified via mirror-image display, where D-protein targets are synthesized for L-peptide binder screening. Once an L-peptide is identified, its mirror-image counterpart will inherently bind to the natural protein, thus yielding a D-peptide binder (Qi et al., 2024; Chang et al., 2015; Zhou et al., 2020). However, the difficulties of D-protein target synthesis limited its application to most real-world drug discovery.

Recently, machine learning-based peptide binder design has been proven practical, and has shown advantages including the specificity to certain epitopes, higher success rate, lower screening cost, and more diverse initial scaffolds for affinity maturation and property optimization (Kong et al., 2025a; Notin et al., 2024). However, current works mainly focus on designing L–L homo-chiral interfaces, while generating D-L hetero-chiral interactions remains largely unexplored. Best to our knowledge, D-Flow (Wu et al., 2024) is the only AI exploration of this topic, which lacks further experimental validation.

In this work, we theoretically analyze the zero-shot cross-chirality generation task for scalarization-based equivariant model (Han et al., 2025). We show that by injecting axial features to polar vector features in geometric neural networks, residues with different chirality will have different but similar latent representations, allowing chirality awareness and cross-chirality generalization. By implementing this in a latent diffusion model, we build PepMirror, the state of the art (SOTA) D-peptide binder *de novo* design model as shown by both *in-silico* and *in-vitro* experiments in wetlabs. In short, our main contributions include:

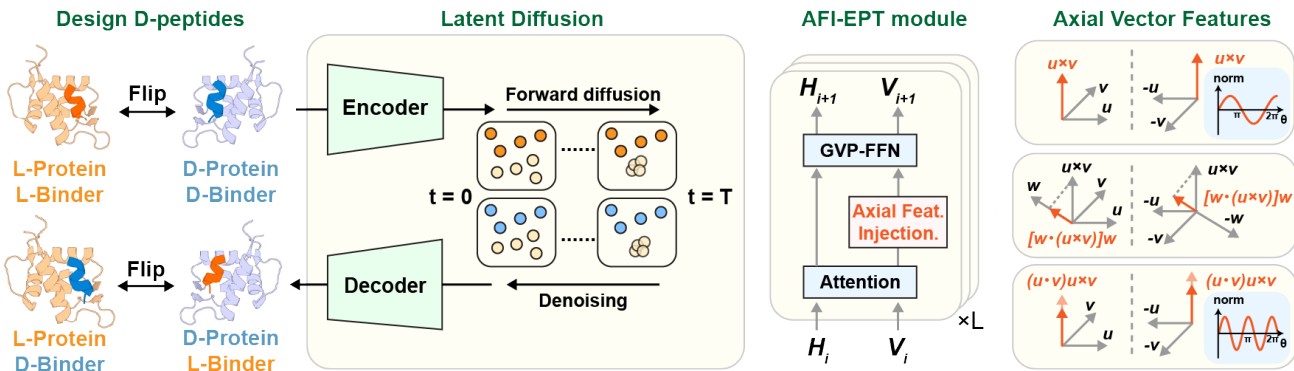

*Figure 1.* We use chiral-sensitive model PepMirror to design D-peptides by flipping. PepMirror is a latent diffusion model using AFI-EPT, which injects axial vector features to learn the chirality. Axial vectors are invariant under the spatial inversion, and we give three direct constructions. The commutator feature (third) captures higher frequency information of the angle between $u, v$.

(1) We propose AFI (Axial Feature Injection), a plug-and-play method to make an $E(3)$-equivariant model to chiral-sensitive (only $SE(3)$-equivariant).

(2) To understand AFI, we provide theoretical analysis together with numerical validation on the AFI-using encoder model by latent space analysis.

(3) Based on AFI, we present PepMirror, an experimentally validated *de novo* framework for mirror-image peptide binder design, supported by comprehensive dry-lab evaluation and wet-lab validation.

## 2. Related Works

**Chirality in protein models** A popular paradigm in protein models is to represent each residue with a local rigid-body frame, parameterized by a translation vector and a rotation matrix from the global frame to the local frame. This method was adopted by many protein structure prediction models such as AlphaFold2 (Jumper et al., 2021), and also many peptide generative models including RFDiffusion (Watson et al., 2023), DiffPepBuilder (Wang et al., 2024), PepFlow (Li et al., 2024), PPFlow (Lin et al., 2024), PepBridge (Li et al., 2025a), D-Flow (Wu et al., 2024), etc. In these models, the chirality of residues is fixed by the definition of the rigid-body and local frames are designed to be right-handed. As a result, a residue and its mirror-image structure will have different rotation matrix but are not in a reflection relationship. In other words, chirality is implicitly encoded as a prior, and the parameterization is reflection-agnostic.

Alternatively, some models parameterize proteins without explicit chirality priors. For instance, PepGLAD (Kong et al., 2025a) and UniMoMo (Kong et al., 2025b) utilize residue-level latent features, FuncBind (Kirchmeyer et al., 2025) employs neural fields to represent atoms as density peaks, PocketXMol (Peng et al., 2025) employs full-atom

diffusion, in which chirality is represented only implicitly through atomic coordinates. As these models typically rely on $E(3)$-equivariant architectures, they are inherently restricted to generating homo-chiral complexes (i.e., L–L or D–D protein–peptide pairs as observed in the training data) unless additional chirality-specific features are introduced. Furthermore, because chirality is not modeled explicitly, the generated peptide ligands frequently exhibit varying degrees of residue-level chirality inversion (Table 1).

**Chirality aware models in geometric learning** There have been multiple approaches to encode chirality of a structure in geometric learning. Representative examples include injecting torsions in message passing (e.g., SphereNet (Coors et al., 2018)) or using coupled torsions and aggregates them with a learnable phase shift to disentangle conformation changes and chirality shifts (e.g., ChIRo (Adams et al., 2021)), designing shift-equivariant yet order-sensitive aggregations (e.g., ChiENN (Gaiński et al., 2023) and Tetra-DMPNN (Pattanaik et al., 2020)) , and introducing parity-even channels such as pseudovectors and mixing them with parity-odd features (e.g., GCPNet (Morehead & Cheng, 2024), REM3DI (Wedig et al., 2025)). Although these architectures have been proved effective for tasks like chirality classification and molecular property prediction, they have not been applied in designing hetero-chiral peptide-protein interfaces.

**Representation theory and chiral features** Equivariant networks grounded in representation theory, such as Tensor Field Networks (Thomas et al., 2018), $SE(3)$-Transformers (Fuchs et al., 2020), and `e3nn` (Geiger & Smidt, 2022), leverage spherical harmonics and tensors to encode rich geometric interactions. While highly expressive (Smidt et al., 2021), these models incur significantly higher computational costs (Li et al., 2025b) compared to standard $E(3)$ Graph Neural Networks (Satorras et al.,

2021). In contrast, we introduce a lightweight axial feature injection that incorporates chirality into efficient $E(3)$ backbones with minimal architectural changes.

**Design D-peptides as L-protein binders** Early efforts on hetero-chiral binder design are mainly based on molecular mechanics, employing methods including the RifGen-RifDock-Rosetta pipeline (Cao et al., 2022; Sun et al., 2024) and fragments assembly approaches (Garton et al., 2018; Engel et al., 2021), which suffer from low success rates, and always need large-scale library screening to identify active molecules. Recently, designing protein-binding proteins based on machine learning has been extensively explored, but few models involve mirror-image peptides. To our knowledge, the only attempt to design D-peptide binders is reported in D-Flow (Wu et al., 2024), where a similar workflow to mirror-image display is utilized.

## 3. Method

### 3.1. Preliminary

A protein can be regarded as a point cloud graph $\mathcal{G}$ of the 3D atom coordinates and atom-types. Inspired by mirror-image display and D-Flow (Wu et al., 2024), a D-peptide binder can be generated by two-step inversion. The L-protein target $\mathcal{G}_t$ can be first inverted to its mirror-image counterpart $\mathcal{G}'_t = P(\mathcal{G}_t)$, where $P$ is the spatial inversion [1] on atoms coordinates: $P = -I_3 \in O(3) \setminus SO(3)$. Then, if the generative model designs an L-peptide binder $\mathcal{G}_b$ against this D-target $\mathcal{G}'_t$, the inverted version of $\mathcal{G}_b$ would be the D-peptide binder for the L-target, and these two binder-target pairs should have the same affinity under $P$. Formally, the desired D-peptide binder is $\mathcal{G}_b = P(f_\theta(P(\mathcal{G}_t)))$, where $f_\theta(x)$ denotes the model parameterized by $\theta$ that generates a peptide binder graph conditioned on $x$.

A popular instantiation of $f_\theta$ is a scalarization-based $E(3)$-equivariant model (Han et al., 2025; Satorras et al., 2021), where a 3D object $X$ is embedded as $\big(H(X), V(X)\big)$, where $H(X) \in \mathbb{R}^{N \times K}$ denotes the $E(3)$-invariant scalar features and $V(X) \in \mathbb{R}^{N \times 3 \times K}$ the $E(3)$-equivariant vector features, $K$ is the number of channels. The neural network updates $(H, V)$ jointly.

### 3.2. Hetero-chiral design as a zero-shot generalization

Due to the homo-chiral feature of native proteins, experimentally resolved structures for hetero-chiral protein–protein interactions are scarce. As a result, peptide generation conditioned on targets of different chirality be-

comes a zero-shot generalization problem: the model is trained on homo-chiral complexes, yet the tasks are under unseen hetero-chiral conditions.

This problem calls for two properties. First, the model must be *chirality-aware*. Second, the representation should remain stable under spatial inversion: for any amino acid $X$, the code of its mirror image $-X$ should stay within the same amino-acid type (differing only by chirality), rather than collapsing or drifting toward other types.

### 3.3. Chirality awareness by introducing axial vectors

To make the equivariant model chirality awareness, we need to break the inversion equivariance of vector features and keep the $SE(3)$-equivariant at the same time.

We introduce *axial vector*, satisfying

$$a(Rx) = \det(R)\, R\, a(x), \quad \forall\, R \in O(3). \qquad (1)$$

Accordingly, *polar vector* is $E(3)$-equivariant. Vector features in vanilla scalarization-based $E(3)$ models are polar vector features. We follow standard terminology in classical physics (Jackson, 2021). Common examples include position and velocity (polar vectors), as opposed to angular momentum and magnetic fields (axial vectors). We propose *Axial Feature Injection* (AFI) by adding axial features to the original polar features by channel wise linear mixing. For $i = 1, \dots, N$, $k = 1, \dots, K$, define the new mixed feature

$$\widetilde{v}_{i,k}(X) := A_k^\top v_{i,:}(X) + B_k^\top a_{i,:}(X), \qquad (2)$$

where $A_k, B_k \in \mathbb{R}^K$ are channel-wise mixing coefficients.

We study the mechanism of AFI by applying an MLP $\varphi$ to the invariant features and the vector norm channel-wise

$$c(X) = \varphi\big(\,[\,H(X), \|\widetilde{V}(X)\|\,]\,\big), \qquad (3)$$

where the norm is computed over 3d spacial dimension, and

$$\widetilde{V}(X) = A^\top v(X) + B^\top a(X) \in \mathbb{R}^{N \times 3 \times K} \qquad (4)$$

is the mixed vector features.

Under some boundedness assumptions on vector features and probability assumptions on mixing parameters, we can prove 3.1 (see the formal version and proof at Theorem B.8 and Corollary B.9).

**Theorem 3.1** (Chirality awareness, informal). *For a sample of amino acid $X$, under some mild assumptions, we have for any $\varepsilon \in (0, 1)$, with probability at least $1 - \delta_W(\varepsilon)$,*

$$\|c(X) - c(-X)\| \geq c_W \varepsilon, \qquad (5)$$

*where $c_W$ is a constant.*

---

[1] In $\mathbb{R}^3$, mirror reflection and spatial inversion are distinct operations, but both have a determinant of $-1$ and differ only by a proper rotation, so they are equivalent for chirality comparison. See Appendix A.1 and B.4 for discussion.

We remark that the existence of a chirality-induced discrepancy is not automatic, even with axial feature injection (AFI). First, if $B = 0$, the model reduces to an $E(3)$-equivariant architecture, hence no discrepancy (cf. Proposition 3.2). Second, even when axial channels are present, certain (non-generic) mixing configurations can eliminate the discrepancy. Take $A = B = I$, and for simplicity we omit the indices $i, k$, then $\widetilde{v}(X) = v(X) + a(X)$ and $\widetilde{v}(-X) = -v(X) + a(X)$. Hence

$$\|\widetilde{v}(-X)\| = \| - v(X) + a(X)\| \tag{6}$$

$$\overset{*}{=} \|v(X) + a(X)\| = \|\widetilde{v}(X)\|, \tag{7}$$

where * holds whenever $v(X) \cdot a(X) = 0$. This orthogonality can indeed occur in structured settings (e.g., when $a$ contains cross-product terms as two cases in our implementation, see Section 3.5). Therefore, one should not expect a uniform deterministic lower bound over all parameter choices. Instead, our theorems establish a *generic* guarantee: the learned parameters fall into the discrepancy-inducing regime with high probability.

Without AFI, the model is equivariant under spatial inversion. By definition, we can prove that absence of AFI implies no discrepancy even for multi-layer models.

**Proposition 3.2** (No discrepancy without AFI). *In the absence of AFI i.e., when using only $E(3)$-equivariant Neural Networks, then we have $c(X) = c(-X)$.*

*Proof.* For $j = 1, \ldots, L$, We denote the features at layer $j$ as $(H_j(X), V_j(X))$. Under the spatial inversion, by the equivariant property, $(H_j(-X), V_j(-X)) = (H_j(X), -V_j(X))$. The scalar output is

$$c(-X) = \varphi([H_L(X), \| - V_L(X)\|]) \tag{8}$$

$$= \varphi([H_L(X), \|V_L(X)\|]) = c(X). \tag{9}$$

Though the chirality awareness is theoretically analyzed only for AFI, we observe a similar nontrivial effect in the full multi-layer model. See Fig. 3 and Section 4.1.

### 3.4. Stable representations for enantiomer structures

**Encoding stability** Although a nontrivial discrepancy is observed under spatial inversion, $c(X)$ and $c(-X)$ represent the same amino-acid type and differ only by chirality. We abstract the multilayer equivariant model as a map $\phi$ and define the latent code by

$$c(X) = \phi\big(H(X), \widetilde{V}(X)\big), \ \widetilde{V}(X) := \{\widetilde{v}_{i,k}(X)\}_{i,k}, \tag{10}$$

where $H(X)$ and $\widetilde{V}(X)$ denote the scalar and vector graph embeddings (see Appendix B.6). To compare molecules with different sizes, we apply zero-padding to align dimensions and define the embedding-space discrepancy

$$d(X_1, X_2) := \|H(X_1) - H(X_2)\| + \|\widetilde{V}(X_1) - \widetilde{V}(X_2)\|. \tag{11}$$

The embedding satisfies $H(-X) = H(X)$, and AFI yields $a(-X) = a(X)$ while $v(-X) = -v(X)$. Hence the change from $X$ to its mirror image $-X$ is confined to the polar contribution, and in particular

$$d(X, -X) = \|\widetilde{V}(X) - \widetilde{V}(-X)\| \tag{12}$$

$$= \|2A\, v(X)\|, \tag{13}$$

where $A, B$ denote the channel-mixing coefficients in Eq. (2).

For an amino acid $X'$ of a different type, both the scalar embedding $H$ and the vector embedding $\widetilde{V}$ are expected to differ substantially, with no symmetry-induced cancellation. We therefore assume the embedding-level separation

$$d(X, X') > d(X, -X). \tag{14}$$

Such assumption can be supported by computing the Tanimoto shape similarity between amino acid pairs, see Fig. 2. While $X$ does not admit a direct correspondence to molecular shape, both are geometry-derived representations. Therefore, the result provides an indirect, geometric corroboration of this assumption.

If $\phi$ preserves this ordering in the sense that larger embedding discrepancies lead to larger code discrepancies, then it follows that

$$\|c(X) - c(-X)\| < \|c(X) - c(X')\|. \tag{15}$$

This provides a simple geometric explanation for the observed 20-*clusters* phenomenon, see Fig. 4 and Section 4.1.

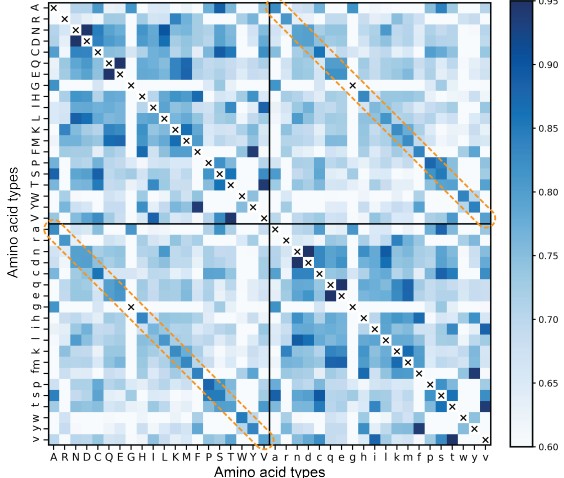

*Figure 2.* The max-pooled pair-wise Tanimoto shape similarity between L/D amino acids. The similarities between an amino acid and its enantiomer are among the highest compared with similaries between different amino acids. Because and similarities between the same amino acid or between "D-Gly" and "L-Gly" are 1.0 by definition, we excluded these entries in the heatmap.

**Diffusion stability**   We establish a continuity theorem on conditional diffusion model. The conditional generation in latent space is an (reversed) SDE sampler:

$$dZ_t = b_\theta(Z_t, t, c)\, dt + \sigma(t)\, dW_t, \qquad t \in [0, T], \quad (16)$$

where $c$ is the encoded code as the condition of diffusion, $b_\theta$ is the learned drift, and $W_t$ is Brownian motion. Let $\mu_c := \mathcal{L}(Z_0 \mid c)$ denote the output distribution at time $t = 0$. We give a Lipschitz assumption on neural networks,

**Assumption 3.3** (Lipschitz drift in state and condition). There exist $L_z, L_c > 0$ such that for all $z, z', c, c', t$,

$$\|b_\theta(z, t, c) - b_\theta(z', t, c)\| \le L_z \|z - z'\|, \quad (17)$$
$$\|b_\theta(z, t, c) - b_\theta(z, t, c')\| \le L_c \|c - c'\|. \quad (18)$$

Under assumption 3.3, using standard coupling method (Levin & Peres, 2017), we can prove (in Appendix B.5)

**Theorem 3.4** (Conditional diffusion is stable in $W_2$). *Under Assumption 3.3, the solutions of diffusion are close in Wasserstein metric,*

$$W_2(\mu_c, \mu_{c'}) \le K_{\text{diff}} \|c - c'\|. \quad (19)$$

*where $K_{\text{diff}}$ is a positive constant.*

It follows that under spatial inversion, the target will give similar latent codes of binders to the decoder. We can expect that this leads to the generalization that the axial-sensitive model could almost always generate L binder given D target even no such pairs in the training data.

### 3.5. Implementation of axial feature injection for D-peptide binder design

Encouraged by the above analysis, we set up to implement AFI within the UniMoMo framework (Kong et al., 2025b), a latent diffusion (Rombach et al., 2022) model that involves a VAE (variational auto-encoder, (Kingma & Welling, 2013)) module and a diffusion module. Briefly, the encoder first maps the input protein structure into a latent code, which is then used to condition the diffusion model. The diffusion generates the latent code of binder, which is decoded into the desired binder using the decoder.

These modules use EPT (Equivariant Pretrained Transformer (Jiao et al., 2024)) as the backbone, which is designed to be $E(3)$-equivariant. EPT then stacks $L$ layers of self-attention and GVP-FFNs (Jing et al., 2020) that preserve the $E(3)$-invariance of $H$ and $E(3)$-equivariance of $V$. To achieve $SE(3)$-equivariance, we modify EPT by adding axial vectors to the vector feature $V_j'$ before every FFN layer. We form *axial vector feature* channels based on polar vector features.

Inspired by the representation theory of $SO(3)$, we leverage irreducible decompositions of tensor products to extract geometric features that encode chirality (Thomas et al., 2018; Geiger & Smidt, 2022). While higher-order tensor representations offer finer geometric resolution, they suffer from high computationaly cost and overfitting to high-frequency noise irrelevant to chirality. We posit that the fundamental parity asymmetry is sufficiently captured by low-order interactions. Therefore, to balance computational efficiency against geometric expressivity, we restrict our framework to decompositions involving up to second-order tensors (see Appendix B.1). Specifically, we construct three axial vector features:

(1) the cross product: $u \times v$

(2) the projection of scalar triple product: $\big(w \cdot (u \times v)\big) \cdot w$

(3) the commutator: $(u \cdot v)(u \times v)$

where $u, v, w \in \mathbb{R}^3$ are adjacent channels of $V' \in \mathbb{R}^{N \times 3 \times K}$. These axial features are then injected via linear mixing of polar and axial channels as shown in Eq. (2). In implementation, we use some normalization on vectors, see Algorithm 1. Finally, we replace $V_j'(X)$ with $\widetilde{V}_j(X)$ in every EPT layer to obtain AFI-EPT, as shown in Figure 1. Direct verification shows that (proved in Appendix B.2)

**Proposition 3.5.** *The constructions of axial vector features above are indeed axial, and the mixed vector features $\widetilde{v}_{i,k}(X)$ are $SE(3)$-equivariant.*

## 4. Experiment

As discussed above, we implement AFI-EPT based on cross products, triple products, and commutators in the framework of UniMoMo (Kong et al., 2025b). Based on the type of axial vector features, we name these models as PepMirror (cross), PepMirror (triple), and PepMirror (commu.). For ablation, we also train the original UniMoMo with linear peptide data for baseline comparison, which is named UniMoMo (pep.). The published version that trained with three modalities (small molecules, peptides and antibodies) is named UniMoMo (all).

### 4.1. Latent space analysis

To assess the effect of AFI and provide numerical evidence for our theoretical discussion, we train an autoencoder with AFI-EPT and analyzed the invariant part of the resulting latent codes for amino acids. Specifically, we apply mean pooling over the atom dimension to obtain a fixed-size representation $c(X) \in \mathbb{R}^8$.

**Discrepancy**   To verify Theorem 3.1 and Proposition 3.2, we analyze the latent codes of each pocket structure $X$ in our LNR test set, obtaining $c(X) \in \mathbb{R}^8$. For each

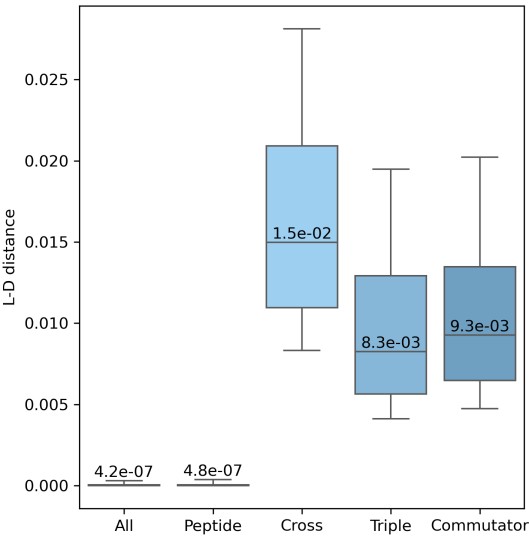

*Figure 3.* Latent-code distances between each amino acid and its inverted counterpart ($X$ vs. $-X$) for L- and D-forms across different encoder variants. All and Peptide refer to UniMoMo without AFI trained on different datasets (see section 4), and the other three are equipped with AFI based on different axial features. Distances are summarized as box plots over all amino acids. Encoders equipped with AFI exhibit a non-negligible inversion-induced discrepancy. The number on every block is the median distance.

sample, we compute the inversion-induced discrepancy $\|c(X) - c(-X)\|_2$ and summarize these distances with box plots (Fig. 3) across samples for several model variants. Introducing AFI consistently increases the L-D discrepancy: the median distance reaches the $10^{-2}$ scale across different axial-feature constructions, more than four orders of magnitude larger than that without AFI.

**Stability and clustering** To compare within-type and between-type separations, we estimate mean latent distances for every pair of amino-acid types. Let $\{X_i\}_{i=1}^{20}$ denote the distributions of the 20 canonical L amino acids, and $\{X_i\}_{i=21}^{40} = \{-X_i\}_{i=1}^{20}$ denote 20 D amino acids. For each $1 \leq i, j \leq 40$, we sample 100 pairs $(x_i, x_j)$ with $x_i \sim X_i$ and $x_j \sim X_j$, and compute

$$\mathbb{E}\big[\|c(x_i) - c(x_j)\|_2\big]. \tag{20}$$

We visualize these means as a heatmap (the right panel of Fig. 4). The three diagonals are about two orders of magnitude smaller than the off-diagonal entries, and the t-SNE plot (the left panel of Fig. 4) can further support the tight within-type clustering. For more visualizations, see Appendix A.2.

## 4.2. In-silico evaluation

### 4.2.1. BASELINES.

We adopt the following peptide design models as baselines, which can be broadly divided into two categories. The first category assumes L chirality as a built-in prior, including RFDiffusion (Watson et al., 2023), DiffPepBuilder (Wang et al., 2024), PepFlow (Li et al., 2024), D-Flow (Wu et al., 2024), PPFlow (Lin et al., 2024), and PepBridge (Li et al., 2025a). These models treat chirality as a preset constraint rather than a learnable or controllable variable.

The second category does not assume a fixed chirality. PocketXMol (Peng et al., 2025) operates directly on atom coordinates, PepGLAD (Kong et al., 2025a) and UniMoMo (Kong et al., 2025b) employs coordinate-derived representations without enforcing chiral constraints. However, this flexibility comes at the cost of limited control over chirality consistency, often resulting in mixed-chirality outputs. Notably, PepGLAD utilizes the idealization method after generation, where generated residues are aligned with ideal templates, ensuring L chirality. PepGLAD with idealization is also tested and reported as PepGLAD(ideal).

### 4.2.2. SETUP AND METRICS

To assess the ability to design D-peptide binders for L-protein targets, we employ large non-redundant complex dataset (LNR) (Tsaban et al., 2022) as our testset. For each model, we input the same binding pocket of either native receptors or the central inverted receptors for L-peptide and D-peptide design, respectively. Generated complexes are then minimized under the Amber14 forcefield (Maier et al., 2015) before the following metrics are calculated:

**Chirality**. We first assess whether generated peptides exhibit the desired chirality. Specifically, we compute the fraction of residues with the correct chirality, where achiral glycines are excluded. We note that some generated structures suffer with severe clashes, leading to chiraliity flip during minimization. We therefore calculate the ratio of desired chirality in both raw outputs and minimized structures. For RFDiffusion and PPFlow that only generate backbone atoms, chirality cannot be directly computed. We therefore set the initial chirality and reconstructed C$\beta$ atoms accordingly based on Ramachandran plots (Appendix A.4).

**Interface affinity**. Because Rosetta energies are statistically derived and exhibit discrepancies between enantiomeric systems (Appendix A.5), we use the score from AutoDock Vina as an alternative metric to assess interface affinity. To evaluate the upper-bound performance of each model, we select the top-1 candidate for each target and report the average binding score. We also compute the proportion of targets for which at least one designed binder achieves a lower interface energy than the native complex, which

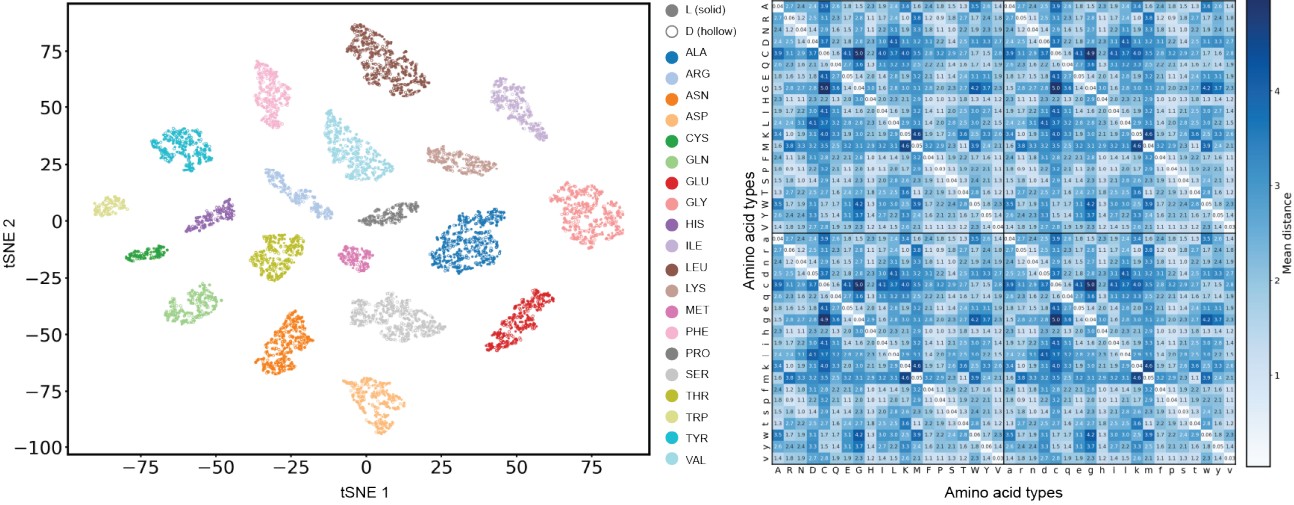

*Figure 4.* Left: t-SNE of 20 types of amino acids including both L and D chirality. As t-SNE cannot keep distance, we plot the heatmap (right) of mean pairwise latent-code distances among 40 amino-acid classes including 20 L amino acids and 20 D amino acids. The three diagonals are two orders of magnitude smaller than the off-diagonal entries ($10^{-2}$ vs 1), indicating tight within-class clustering and clear inter-class separation. The model we use here is PepMirror (cross), for other models, see Fig. S4, Fig. S5, and Fig. S6

we report as the interface energy improvement (IMP). To characterize overall performance, we additionally report the mean binding energy across all candidates with negative binding energies. Besides, the ratio of complexes with negative binding energies is reported as the success rate (Suc.).

### 4.2.3. RESULTS

**Chirality**. Table 1 reveals two types of behaviours. Frame-based models implicitly enforce L chirality, whereas models without an explicit chirality prior generate mixed-chiral peptides. Notably, PocketXMol, PepGLAD and UniMoMo show $E(3)$-equivariance: inverting the input structure leads to an accordingly inverted output, which is counterproductive for hetero-chiral binder design. This equivariance is reflected by the observation that the L- and D-correct chirality fractions approximately sum to 100%.

After minimization, most structures preserve their initial chirality. For PepBridge, severe intra-ligand clashes lead to loss of chiral consistancy during relaxation. Among baselines, PepMirror achieves the highest chirality consistency, and also designs reasonable backbone Ramachadran torsions (Figure S7), suggesting that AFI effectively yields an $SE(3)$-equivariant mapping that supports hetero-chiral binder design.

**Interface affinity**. To align with downstream applications where D-peptides are valued for their consistent D-chirality and thus guaranteed stability, we exclude methods that cannot reliably generate peptides with consistent residue-level chirality from the following evaluations.

*Table 1.* Chirality and minimization backbone-RMSD of models on L/D-peptide design tasks. The best and second best data is labeled orange/light orange for L tasks, and blue/light blue for D tasks.

| Models | Task | Right Chirality% | |
|---|---|---|---|
| | | Raw | Minimized |
| RFDiffusion | L | 100.0 | 99.57 |
| | D | 100.0 | 99.15 |
| PPFlow | L | 100.0 | 95.90 |
| | D | 100.0 | 96.31 |
| DiffPepBuilder | L | 100.0 | 98.04 |
| | D | 100.0 | 97.91 |
| PepFlow | L | 100.0 | 99.06 |
| | D | 100.0 | 98.99 |
| D-Flow | L | 100.0 | 98.43 |
| | D | 100.0 | 98.54 |
| Pepbridge | L | 100.0 | 60.04 |
| | D | 100.0 | 60.39 |
| PepGLAD(ideal) | L | 100.0 | 99.01 |
| | D | 100.0 | 99.04 |
| PocketXMol | L | 57.83 | 57.83 |
| | D | 43.12 | 43.12 |
| PepGLAD | L | 50.10 | 50.28 |
| | D | 49.94 | 49.85 |
| UniMoMo(pep.) | L | 77.03 | 76.98 |
| | D | 23.90 | 23.95 |
| UniMoMo(all) | L | 84.78 | 84.70 |
| | D | 15.70 | 15.76 |
| PepMirror(cross) | L | 99.93 | 99.83 |
| | D | 99.91 | 99.81 |
| PepMirror(triple) | L | 99.86 | 99.75 |
| | D | 99.84 | 99.75 |
| PepMirror(commu.) | L | 99.95 | 99.88 |
| | D | 99.94 | 99.88 |

Among the remaining baselines, RFDiffusion shows a pronounced performance cliff from L- to D-peptide design: the average affinity decreases from -3.30 to -1.77, and IMP drops from 44.09% to 16.13%. Other methods exhibit a similar L–D gap, most clearly in success rate (Table 2). Together with the substantially higher structural diversity observed on D-peptide tasks (Table S5), these results suggest that many models explore a broader yet less target-aligned conformational space for D-peptide design.

PepMirror achieves the strongest overall performance, which is insensitive to the choice of axial vector. It also exhibits the smallest L-to-D degradation, yielding a larger relative advantage on D-peptide tasks. Besides Vina score, we also assess interface quality using Rosetta ddG, while we note its inconsistency on protein enantiomers. Under this metric, PepMirror remains the top performer and shows a larger advantage among baselines, further supporting its ability to design high-quality hetero-chiral interfaces (Appendix A.5 and Table S7).

### 4.3. Wet-lab validation

Motivated by PepMirror's clear advantages in *in-silico* evaluations, we next assess its practical utility for *de novo* D-peptide binder discovery. Using PepMirror (cross), we generate 5,000 D-peptide candidates against CD38 (Cluster of Differentiation 38), a validated therapeutic target in multiple myeloma and an emerging target in NAD$^+$-linked immunometabolic disorders. After physics-based and geometry-based filtering (Appendix C.6), we prioritize 12 candidates for chemical synthesis and binding assays. Among them, a 10-mer peptide (D-1412; sequence "trikhytyce") achieve a dissociation constant $K_D \approx 10\ \mu M$, with kinetic and steady-state fittings yielding consistent estimates. Structural inspection of D-1412 suggests multiple sidechain-mediated interactions that depends on correct stereochemistry, supporting PepMirror's capability to design plausible hetero-chiral interactions (Figure 5). However, we observed an unexpected phenomenon: the enantiomer of D-1412 also showed binding activity toward CD38 with a comparable affinity (Appendix A.6). While peptide–protein interactions are generally considered stereoselective, recent studies suggest that this selectivity is not necessarily absolute and may be attenuated by conformational disorder or alternative binding modes (Newcombe et al., 2024; Li et al., 2026). To validate this observation, we repeated the BLI assays and confirmed peptide chirality by circular dichroism (CD) (Figure S8). These controls support the reliability of the affinity measurements for both peptides. Overall, these results imply a competitive experimental hit rate and further demonstrate the practical utility of PepMirror for mirror-image drug discovery.

## 5. Conclusion and Discussion

In this work, we propose AFI-EPT that injects axial vector features into polar vector features in EPT (Jiao et al., 2024). By implementing this module in latent diffusion framework, we build PepMirror that design mirror-image peptide binders for native protein targets. Through theoretical analysis and experiments, we show that the latent codes of L and D amino acids will have close but different representations, so that the model acquires the ability to distinguish different chirality while maintains the ability to generate reasonable structures given unseen D-targets as input. The evaluation results show that PepMirror have advanced performance compared with existing peptide binder design models. On top of this, we tested PepMirror in a real-world D-peptide binder design campaign, and successfully identified a D-binder against CD38 with a KD of 10 $\mu M$ out of 12 designs.

Although PepMirror has achieved best-in-class performance and, for the first time, demonstrate utility in wet-lab experiments, we recognize that opportunities remain for further exploration. First, our theoretical analysis of AFI focuses on the feature-mixing mechanism under simplifying assumptions, rather than providing an end-to-end theory of the trained network (including optimization and generalization). A more complete account of how axial information propagates through subsequent equivariant blocks remains an open direction.

Besides, although the main experiments focus on three simple axial-vector constructions, AFI should be viewed more broadly as a lightweight design principle that is plug-and-

*Table 2.* Interface quality of of generated L/D-peptides by different models. The best and second best data is labeled orange/light orange for L tasks, and blue/light blue for D tasks.

| Models | Task | Suc.% | Avg. | Top | IMP% |
|---|---|---|---|---|---|
| RFDiffusion | L | 99.52 | -3.30 | -5.14 | 44.09 |
| | D | 98.58 | -1.77 | -3.78 | 16.13 |
| DiffPepBuilder | L | 86.56 | -3.58 | -5.44 | 56.99 |
| | D | 80.14 | -3.38 | -5.23 | 50.54 |
| PepFlow | L | 99.41 | -3.31 | -4.36 | 13.98 |
| | D | 96.78 | -2.75 | -4.15 | 16.13 |
| D-Flow | L | 99.31 | -3.33 | -4.41 | 10.75 |
| | D | 97.52 | -3.11 | -4.54 | 22.58 |
| PPFlow | L | 65.94 | -2.58 | -5.09 | 40.86 |
| | D | 64.46 | -2.64 | -5.21 | 48.39 |
| PepGLAD(ideal) | L | 94.56 | -3.27 | -5.10 | 40.86 |
| | D | 95.08 | -3.26 | -5.11 | 43.01 |
| PepMirror(cross) | L | 99.67 | -4.27 | -5.81 | 69.89 |
| | D | 99.76 | -4.15 | -5.69 | 63.44 |
| PepMirror(triple) | L | 99.73 | -4.31 | -5.88 | 69.89 |
| | D | 99.72 | -4.20 | -5.75 | 67.74 |
| PepMirror(commu.) | L | 99.73 | -4.34 | -5.89 | 76.34 |
| | D | 99.75 | -4.25 | -5.87 | 72.04 |

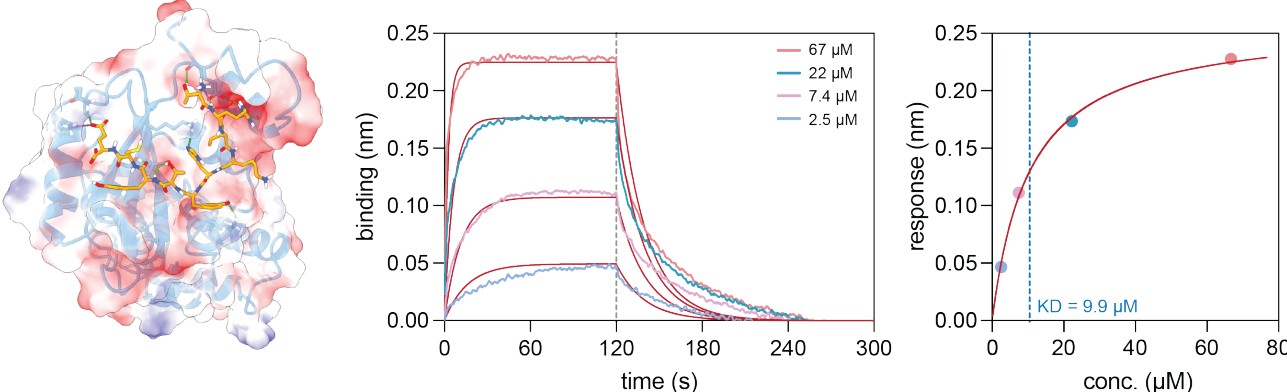

*Figure 5.* The identified D-peptide binder against CD38. Left: Complex structure of D-1412 and CD38 generated by PepMirror (cross), where multiple interactions can be identified. Middle: Stacked curves of association and dissociation under different concentrations with kinetic fitting. Right: Steady state fitting of the max response for each concentration, the blue line is the observed KD.

play for many model architectures. As preliminary evidence, we evaluated a mixed cross–triple–commutator variant and a pseudo-scalar injection variant of PepMirror (Appendix A.7). These extensions show similar behavior with the AFI variants in our main context, suggesting that AFI is transferable across reasonable implementation choices and may serve as a general strategy for introducing chirality sensitivity.

Overall, our work shows the feasibility of designing wet-lab validated mirror-image peptide binders with generative AI for the first time, which not only provides a useful tool, but also inspires new insights in handling chirality in protein design.

## Acknowledgements

We sincerely thank our reviewers for their valuable discussions and comments, as well as Xiangzhe Kong, Mingyu Li, Ziting Zhang, and other colleagues in Anew Labs for their inspiring advice and help. We would also like to thank Innovative Drug Research and Development–National Science and Technology Major Project (No.2025ZD1802501); Beijing Frontier Research Center for Biological Structure Fundings; the National Key R&D Program of China (2022YFC3401500); the National Natural Science Foundation of China (T2488301, 22227810, and 22137005); Fundamental and Interdisciplinary Disciplines Breakthrough Plan of the Ministry of Education of China (JYB2025XDXM501); the National Facility for Translational Medicine(Shanghai) Fundings; the Fundamental Research Funds for the Central Universities; Tsinghua-Peking Center for Life Sciences; Ministry of Education Key Laboratory of Bioorganic Phosphorus Chemistry and Chemical-Biology; Center for Synthetic and Systems Biology, Tsinghua University; the XPLORER prize; the New Cornerstone Science Foundation; and AI Industry Research Innovation Center, Wuxi Research Institute for Applied Technologies, Tsinghua University for their support.

## Impact Statement

This paper presents work whose goal is to advance the field of machine learning based functional protein design. There are many potential societal consequences of our work, none of which we feel must be specifically highlighted here.

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

# A. Additional Results and Discussion

## A.1. On the chirality conversion and spacial inversion

The commonly used definition of chirality is the property that a structure cannot be superimposed with its mirror-image (i.e., the reflected structure). In this paper, we used central inversion as the operation that converses chirality, because mirror reflection and central inversion both have a determinant of -1 and differ only by a proper rotation, so they are equivalent for chirality comparison. We use central inversion because it flips all three coordinates simultaneously and thus treats them uniformly, whereas a mirror reflection requires choosing a specific mirror plane. Using central inversion for chirality conversion has also been reported in D-Flow (Wu et al., 2024).

To experimentally validate the equivalency, we reflected the structures in the LNR test set across the xy, yz, and xz planes. In addition, we randomly sampled three other planes passing through the structural center and reflected the structures across them (denoted as random_1/2/3). We evaluated PepMirror(cross) in terms of chirality and interface energy with these six variants of LNR as targets. The result shows highly similar performance across different reflection planes, supporting that reflections across different planes do not affect the generation quality (Table S1).

*Table S1.* Comparing model performances on test sets after inversion and reflection based on different planes

| Operation | Chirality | Right Chirality% | | Vina Score | | | |
|---|---|---|---|---|---|---|---|
| | | Raw | Minimized | Suc.% | Avg. | Top | IMP% |
| original | L | 99.93 | 99.83 | 99.67 | -4.27 | -5.81 | 69.89 |
| inversion | D | 99.91 | 99.81 | 99.76 | -4.15 | -5.69 | 63.44 |
| reflection_xy | D | 99.90 | 99.82 | 99.70 | -4.18 | -5.72 | 65.59 |
| reflection_yz | D | 99.92 | 99.83 | 99.62 | -4.18 | -5.73 | 69.89 |
| reflection_xz | D | 99.90 | 99.82 | 99.70 | -4.19 | -5.72 | 65.59 |
| random_1 | D | 99.90 | 99.83 | 99.71 | -4.19 | -5.72 | 66.67 |
| random_2 | D | 99.91 | 99.81 | 99.59 | -4.19 | -5.74 | 65.59 |
| random_3 | D | 99.92 | 99.83 | 99.70 | -4.19 | -5.70 | 65.59 |

Moreover, we evaluated equivariance at the representation level by comparing residue-level latents of the original, inverted, and reflected LNR pockets. The results show that reflections across different planes still separate L/D amino-acids effectively, while preserving consistent residue latents under rotation (Table S2).

*Table S2.* Median distances of pocket latents under different chirality operations relative to those under original and inversion

| Operation | Chirality | Med. Dist. to original | Med. Dist. to inversion |
|---|---|---|---|
| original | L | 0 | 1.5e-2 |
| inversion | D | 1.5e-2 | 0 |
| reflection_xy | D | 1.5e-2 | 5.3e-7 |
| reflection_yz | D | 1.5e-2 | 5.6e-7 |
| reflection_xz | D | 1.5e-2 | 3.9e-7 |
| random_1 | D | 1.5e-2 | 3.0e-5 |
| random_2 | D | 1.5e-2 | 3.1e-5 |
| random_3 | D | 1.5e-2 | 3.1e-5 |

We noted that randomly sampled planes show larger deviations from inversion than axis-aligned reflections. We believe this mainly comes from the limited precision of the PDB format, where coordinates are typically stored with three decimal places. Arbitrary-plane reflections are therefore accumulate larger rounding errors, whereas axis-aligned reflections are essentially sign flips and are numerically more stable. Since PDB is commonly used for protein structures, we view this as a realistic setting. Under this precision, AFI preserves rotation equivariance and maintains chirality separation under arbitrary reflections.

## A.2. Latent space analysis in detail

We summarize the PCA, t-SNE, and UMAP visualizations, together with the pairwise distance heatmaps, for the three axial features used in AFI-EPT. Figures S1, S4, S5,and S6 show that all three features organize amino-acid embeddings into 20 well-separated clusters in latent space, corresponding to the 20 canonical residue types, with L/D isomers of the

same residue co-clustering. The distance heatmaps across residue types further corroborate this 20-cluster structure. The parameters and explained variance ratios are listed in Table S3 and Table S4.

We also show the zoom-in version of the tSNE latent clusters in Figure S2 and S3, where the latent codes of L/D chirality for each amino acid type are plotted separately.

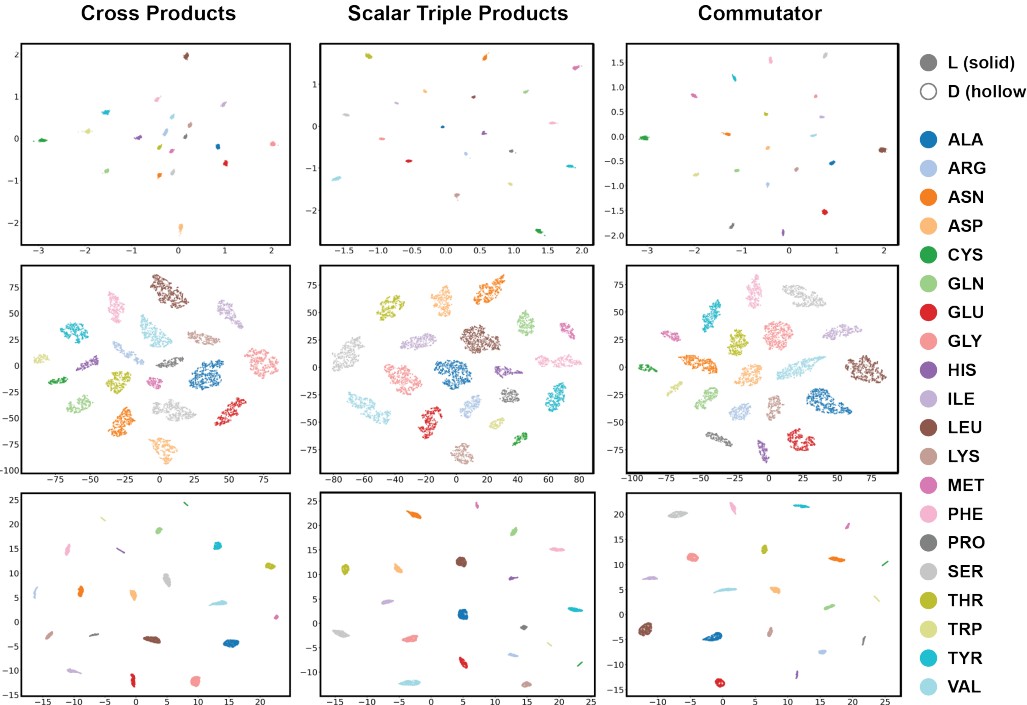

*Figure S1.* visualization of the clustering results of the LNR latent codes generated from PepMirror. From top to bottom, the rows display the clustering results using PCA, t-SNE, and UMAP, respectively.

*Table S3.* Parameters used in latent clustering

| Method | Parameter | Value |
|---|---|---|
| PCA | random state | 12 |
| tSNE | perplexity | 50.0 |
| | random state | 12 |
| | metric | euclidean |
| UMAP | n neighbors | 75 |
| | random state | 12 |
| | min dist | 0 |

*Table S4.* Explained variance ratios in PCA

| Cross | | Triple | | Commutator | |
|---|---|---|---|---|---|
| PC | ratio | PC | ratio | PC | ratio |
| 1 | 0.346 | 1 | 0.523 | 1 | 0.547 |
| 2 | 0.342 | 2 | 0.477 | 2 | 0.453 |
| 3 | 0.312 | 3 | 3.4e-5 | 3 | 2.5e-5 |
| 4 | 4.9e-5 | 4 | 2.2e-5 | 4 | 2.1e-5 |
| 5 | 1.7e-5 | 5 | 9.9e-6 | 5 | 1.2e-5 |
| 6 | 1.2e-5 | 6 | 8.0e-6 | 6 | 6.9e-6 |
| 7 | 6.8e-6 | 7 | 7.3e-6 | 7 | 4.3e-6 |
| 8 | 4.1e-6 | 8 | 5.3e-6 | 8 | 2.7e-6 |

### A.3. Evaluations on native interface coverage and diversity

**Settings** **For native interface recovery**. Although D-peptide binders are not expected to match native binder sequences, native complexes provide a useful reference for plausible binding poses. We computed binding site recovery (BSR) to quantify how well the designed interface overlaps with the native binding site, capturing both the contact coverage and the epitope precision. **For diversity**. Generating diverse candidates provides more starting points for downstream optimization. We quantified diversity at both the sequence and structure levels by clustering generated peptides using sequence overlap and C$\alpha$ RMSD thresholds, respectively. Diversity is defined as $N_{\text{clusters}}/N_{\text{samples}}$, reported as $Div_{\text{Seq}}$ and $Div_{\text{Struct}}$.

**Results and analysis** Interface recovery and diversity pull the design in different directions, reflecting the trade-off between fidelity to the native binding mode and exploration for alternative conformations that could be better solutions.

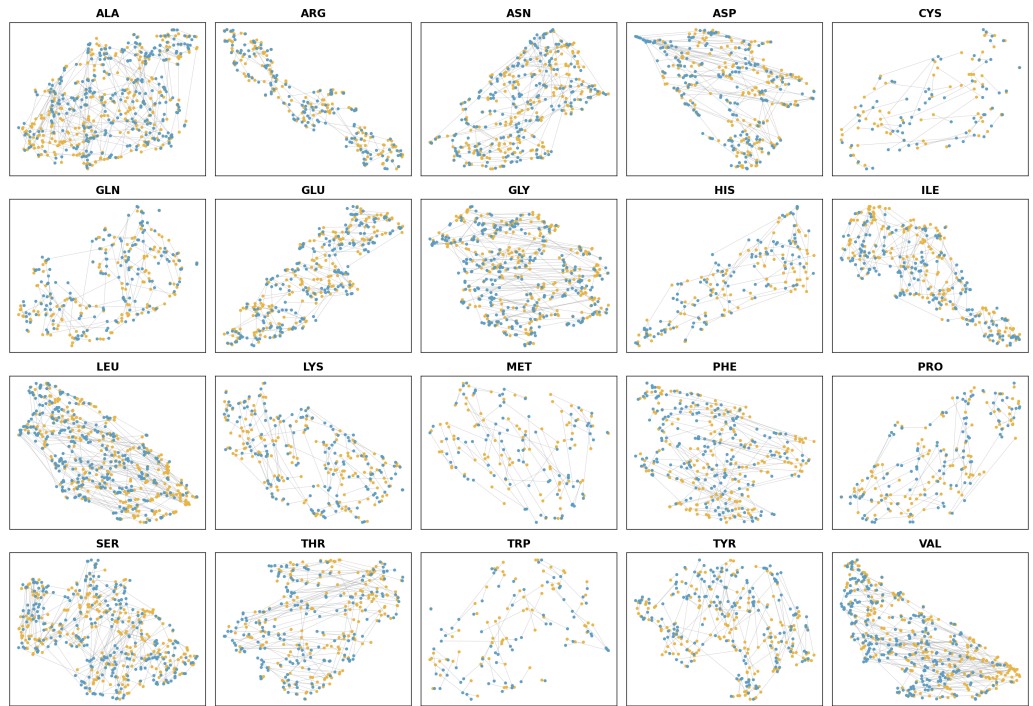

*Figure S2.* Visualization of the tSNE clustering results of the LNR latent codes generated from PepMirror(cross). Latent codes for L amino acids are orange, latent codes for D amino acids are blue, and a residue and its inverted structure are connected with a gray line.

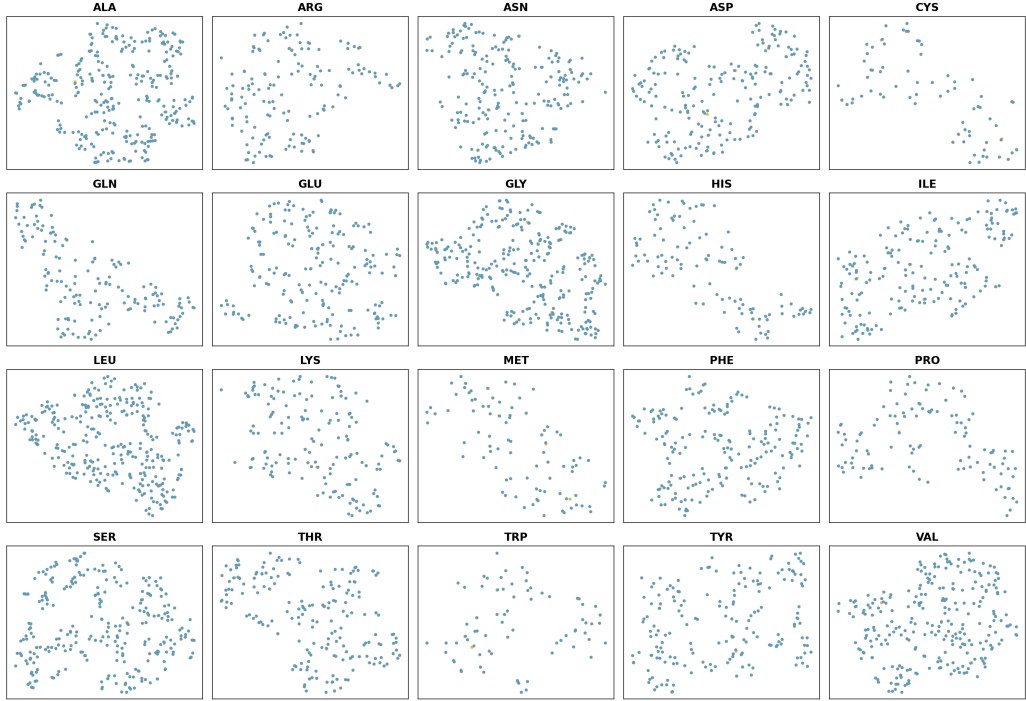

*Figure S3.* Visualization of the tSNE clustering results of the LNR latent codes generated from UniMoMo(pep.). Latent codes for L amino acids are orange, latent codes for D amino acids are blue, and a residue and its inverted structure are connected with a gray line.

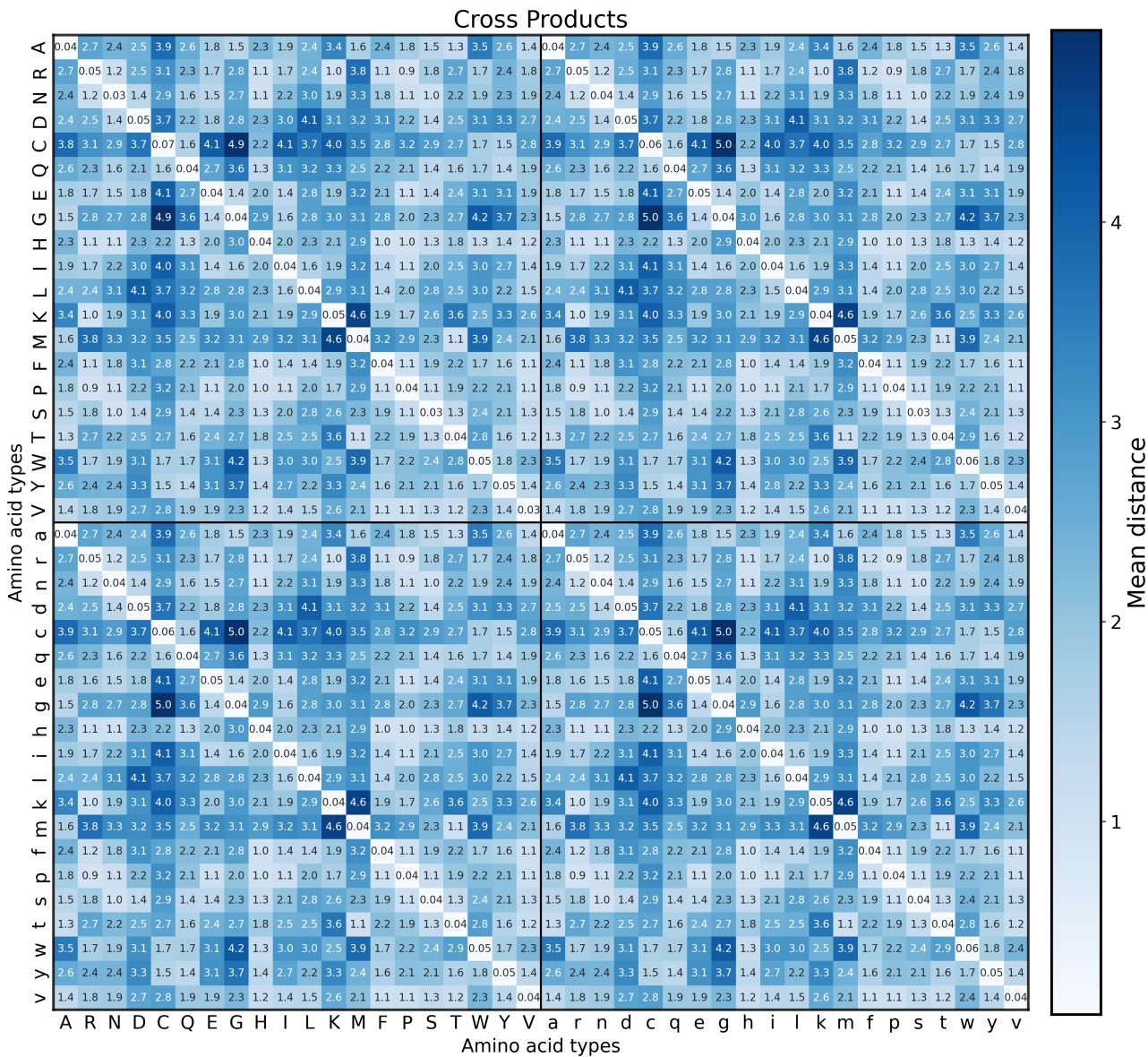

*Figure S4.* Heatmaps of mean-pooled euclidean distances between each amino acid type pair. The encoder of PepMirror(cross) is employed for encoding.

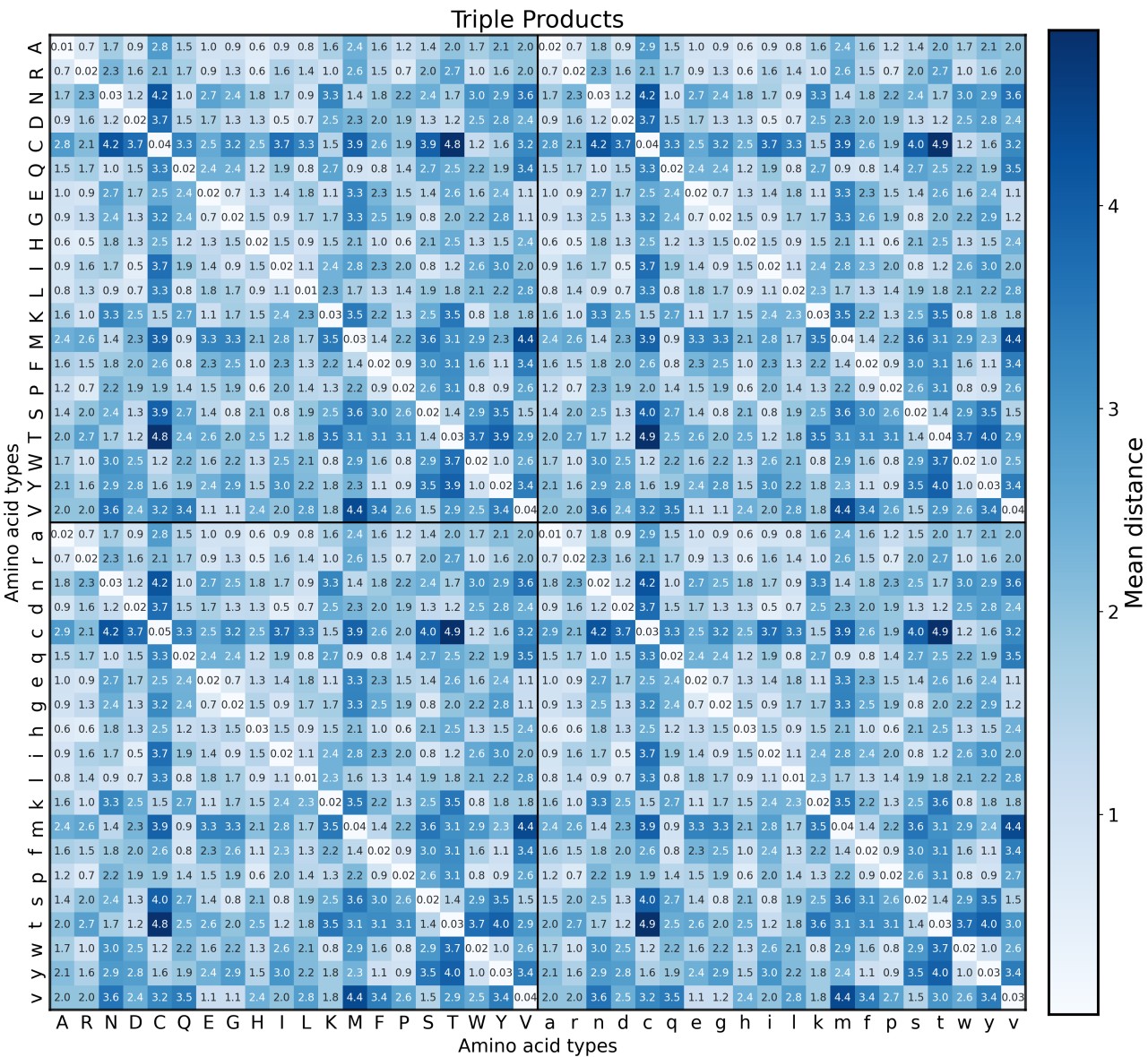

*Figure S5.* Heatmaps of mean-pooled euclidean distances between each amino acid type pair. The encoder of PepMirror(triple) is employed for encoding.

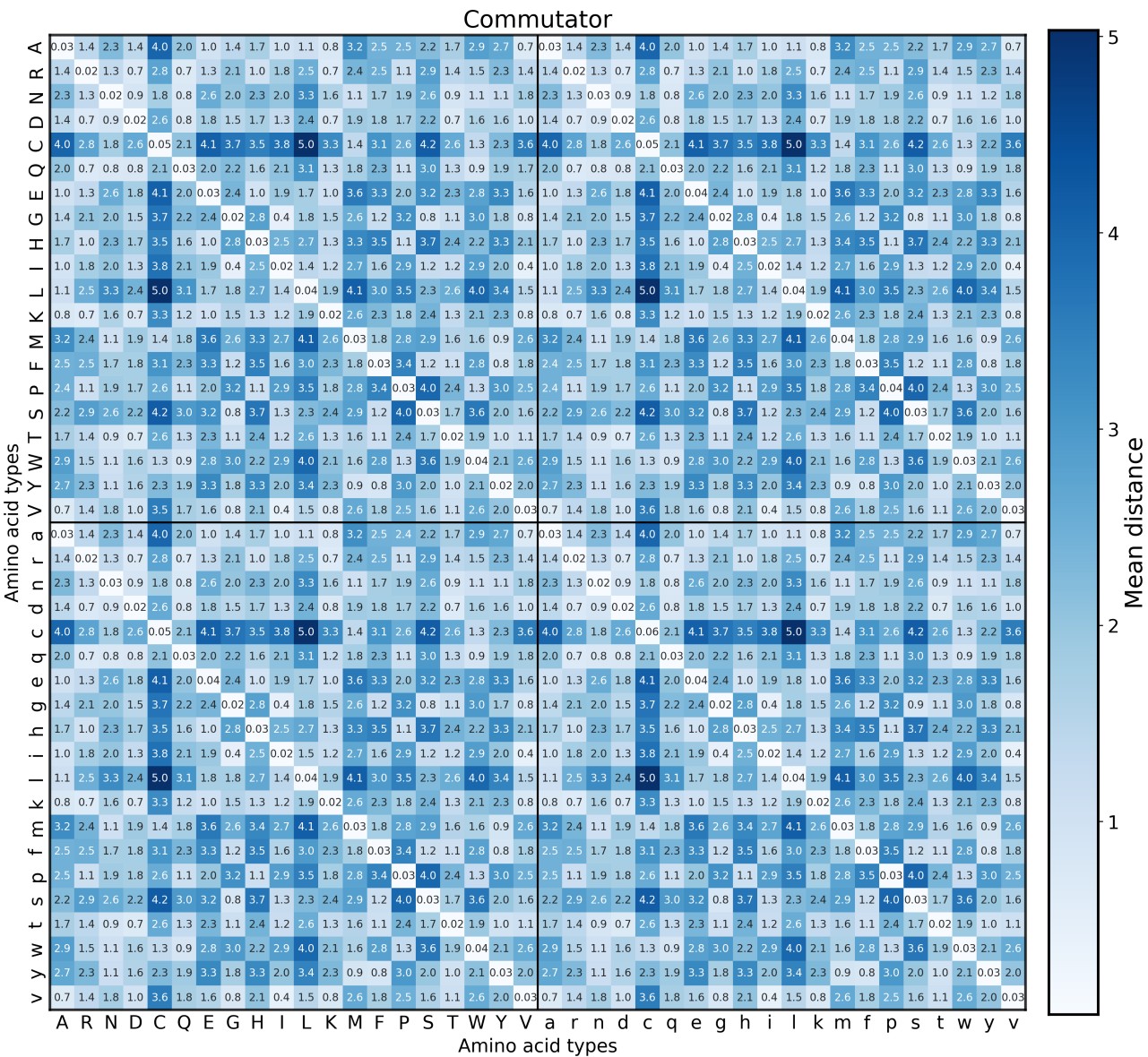

*Figure S6.* Heatmaps of mean-pooled euclidean distances between each amino acid type pair. The encoder of PepMirror(commutator) is employed for encoding.

*Table S5.* Native interface coverage and diversity of generated L/D peptides by different models. The highest and the second highest value of each metric are labeled with orange/light orange for L tasks, and blue/light blue for D tasks.

| Models | Task | BSR | $Div_{\text{seq}}$ | $Div_{\text{struct}}$ |
|---|---|---|---|---|
| RFDiffusion | L | 62.39 | 0.496 | 0.701 |
| | D | 35.53 | 0.271 | 0.934 |
| DiffPepBuilder | L | 90.24 | 0.211 | 0.319 |
| | D | 89.63 | 0.237 | 0.383 |
| PepFlow | L | 82.33 | 0.095 | 0.172 |
| | D | 73.57 | 0.145 | 0.346 |
| D-Flow | L | 81.48 | 0.063 | 0.355 |
| | D | 79.93 | 0.082 | 0.446 |
| PPFlow | L | 82.01 | 0.856 | 0.916 |
| | D | 81.78 | 0.859 | 0.920 |
| PepGLAD(ideal) | L | 78.05 | 0.860 | 0.868 |
| | D | 78.24 | 0.861 | 0.867 |
| PepMirror(cross) | L | 87.51 | 0.846 | 0.719 |
| | D | 86.64 | 0.847 | 0.717 |
| PepMirror(triple) | L | 87.19 | 0.840 | 0.682 |
| | D | 86.52 | 0.848 | 0.727 |
| PepMirror(commu.) | L | 88.12 | 0.839 | 0.648 |
| | D | 87.34 | 0.848 | 0.686 |

Therefore, although these metrics are often treated as "the higher, the better", excessively large values in either can be undesirable in practice. Overly high recovery may indicate limited exploration, whereas overly high diversity can signal weak adherence to the specified binding mode.

For most models, generated peptides cover more than 50% of the native epitope, with DiffPepBuilder showing the highest coverage at around 90%. This proximity to native states likely contributes to the competitive average and top interface energy in Table 2, where DiffPepBuilder nearly matches PepMirror. However, such adherence restricts sequence and structural diversity, leading to a markedly lower IMP than PepMirror despite similar affinity.

In contrast, RFDiffusion shows substantially lower coverage, especially on D-peptide design tasks. Consistently, it exhibits high D-peptide structural diversity (0.934), which largely stems from limited site specificity: many designs drift from the designated hotspots rather than diversifying within the intended pocket. PPFlow shows high diversity in both sequence and structure, likely due to weaker pocket-shape conditioning and limited clash control: peptides are less constrained by the intended geometry and often sample a broader, clash-prone conformational space. Unlike RFDiffusion, which frequently drifts away from the target region, PPFlow remains more engaged with the receptor, inducing more complex interfaces and consequently higher sequence diversity.

By comparison, PepMirror maintains BSR and diversity within a reasonable range, with only a small gap between L- and D-peptide tasks. This balance between reliable epitope anchoring and principled exploration may explain its strong IMP (Table 2 and Table S7).

**A.4. Ramachandran plot analysis**

The backbone conformation of a residue can be characterized by two torsions, known as the Ramachandran angles. Specifically, $\phi$ is the dihedral angle around the N–C$_\alpha$ bond defined by the four atoms $(C_{i-1}, N_i, C_{\alpha i}, C_i)$, and $\psi$ is the dihedral angle around the C$_\alpha$–C bond defined by $(N_i, C_{\alpha i}, C_i, N_{i+1})$. Statistical analyses on known protein structures have shown that there is a preferred area in the joint distribution of $(\phi, \psi)$, which depends on side-chain structures. In other words, allowed conformations concentrate in certain regions of the $\phi$–$\psi$ plane. By definition, the Ramachandran distribution of a protein is the central inversion of that of its mirror-imgae, i.e., $(\phi, \psi) \mapsto (-\phi, -\psi)$. Therefore, Ramachandran plots provide a diagnostic for: (i) the physical plausibility of peptide backbones, and (ii) whether the sampled torsional preferences are consistent with the intended residue chirality.

Figure S7 reports the Ramachandran plots of peptides generated by some models. For RFDiffusion and PPFlow in line 1, we show that the backbone of generated peptides are align to be L-residues no matter what chirality the receptor is. In line 2, we show that although idealization could ensure homo L-chirality for PepGLAD, the mainchain torsions are still not ideal

*Table S6.* The energy discrepancy of protein enantiomers calculated by Rosetta

| Entry | Backbone | | Sidechain | | Jump | Round | Score Function | $\Delta$Total Energy | $\Delta$ddG | $\Delta$dG |
| | Lig. | Rec. | Lig. | Rec. | | | | | | |
|---|---|---|---|---|---|---|---|---|---|---|
| 1 | True | False | True | True | False | 2 | ref2015 | 5098.10 | 19.95 | 16.61 |
| 2 | True | True | True | True | False | 2 | ref2015 | 2772.00 | 13.25 | 12.03 |
| 3 | True | True | True | True | True | 2 | ref2015 | 2747.82 | 12.27 | 11.49 |
| 4 | True | True | True | True | False | 5 | ref2015 | 2757.52 | 13.09 | 11.81 |
| 5 | True | True | True | True | False | 2 | beta_nov16 | 2469.61 | 13.34 | 12.05 |

for L-peptides. In line 3, we show that UniMoMo(all) with original EPT has the E(3)-equivariant feature, where inverting targets causes the inversion of Ramachadran plots. In contrary, PepMirror with AFI-EPT not only maintains chirality at residue level, but also keeps backbone dihedrals to be suitable for L-peptides.

## A.5. Interface affinity analysis by Rosetta

**Discrepancy of Rosetta score between protein enantiomers**. To compare interface energies between L–L and L–D complexes reliably, the scoring function must be $E(3)$-*invariant*: a structure and its enantiomer should receive the same total energy, and a complex should have the same binding (interface) energy as its mirror image.

Although Rosetta technically supports D-amino-acid residues, we empirically observe substantial inconsistencies when scoring enantiomeric inputs. Specifically, when we provide complexes from the LNR dataset and their mirror images to Rosetta and compute both total and interface energies, the resulting scores differ markedly. This discrepancy persists across multiple score functions and remains even after varying relaxation protocols, indicating that the lack of enantiomer consistency is not easily mitigated by choices of score functions or relax settings (Table. S6).

A detailed breakdown of the score terms reveals that the discrepancy is dominated by an abnormal increase in `fa_rep` after spatial inversion. We hypothesize that this behavior stems from Rosetta's discrete, rotamer-library–based sampling. Because statistical coverage for D-protein conformations is limited, conformers that are deemed permissible for L-proteins may become underrepresented for their D counterparts. As a result, the relax trajectory can be biased, leading to higher steric repulsion. In contrast, full-atom forcefield–based tools such as AutoDock Vina yield nearly identical scores for enantiomeric protein complexes (the average score for L and D are both -4.57). We therefore adopt Vina as our interface-affinity evaluation metric. Nonetheless, we still report the results of interface affinity evaluation based on Rosetta for references.

**Interface affinity evaluation based on Rosetta**.

We follow the evaluation protocol in entry 1 of Table S6 to compute Rosetta ddG. Analogous to the interface affinity metrics in Section 4.2.2, we report (i) top-1 ddG across multiple samples and the corresponding interface energy improvement (IMP) to capture best-case performance, and (ii) the success rate (fraction of designs with ddG< 0) and the mean ddG over successful designs to reflect overall quality. For consistency, IMP is referenced to ddG values computed on L–L LNR complexes, while we note that directly comparing D–L to L–L may be biased due to Rosetta's chirality-dependent discrepancy.

Table S7 exhibits trends consistent with Table 2. In particular, the L–D performance cliff becomes more pronounced under Rosetta ddG. Some baselines achieve performance comparable to PepMirror on L-peptide tasks, yet suffer substantial degradations across metrics on D-peptide tasks. Consequently, PepMirror shows a markedly larger advantage on D-peptide design in this evaluation.

## A.6. Stereo-selectivity of the designed peptide binder

After identifying D-1412 as a binder towards CD38, we synthesized and tested its enantiomer (named L-1412) in terms of the CD38 binding affinity. The BLI result shows a comparable affinity of L-1412 as well, indicating the lack of stereo-selectivity of D-1412. This result aligns with recent evidence that shows enantiomers can both retain binding, and the affinity difference decreases as the structure gets more disordered (Newcombe et al., 2024). And some enantiomer pairs that both have binding affinity may not have the same binding area (Li et al., 2026). Considering D-1412 does not have a rigid folding structure, the reduced stereo-selectivity is understandable. We also confirmed the chirality of the enantiomers by cCircular dichroism (Figure S8).

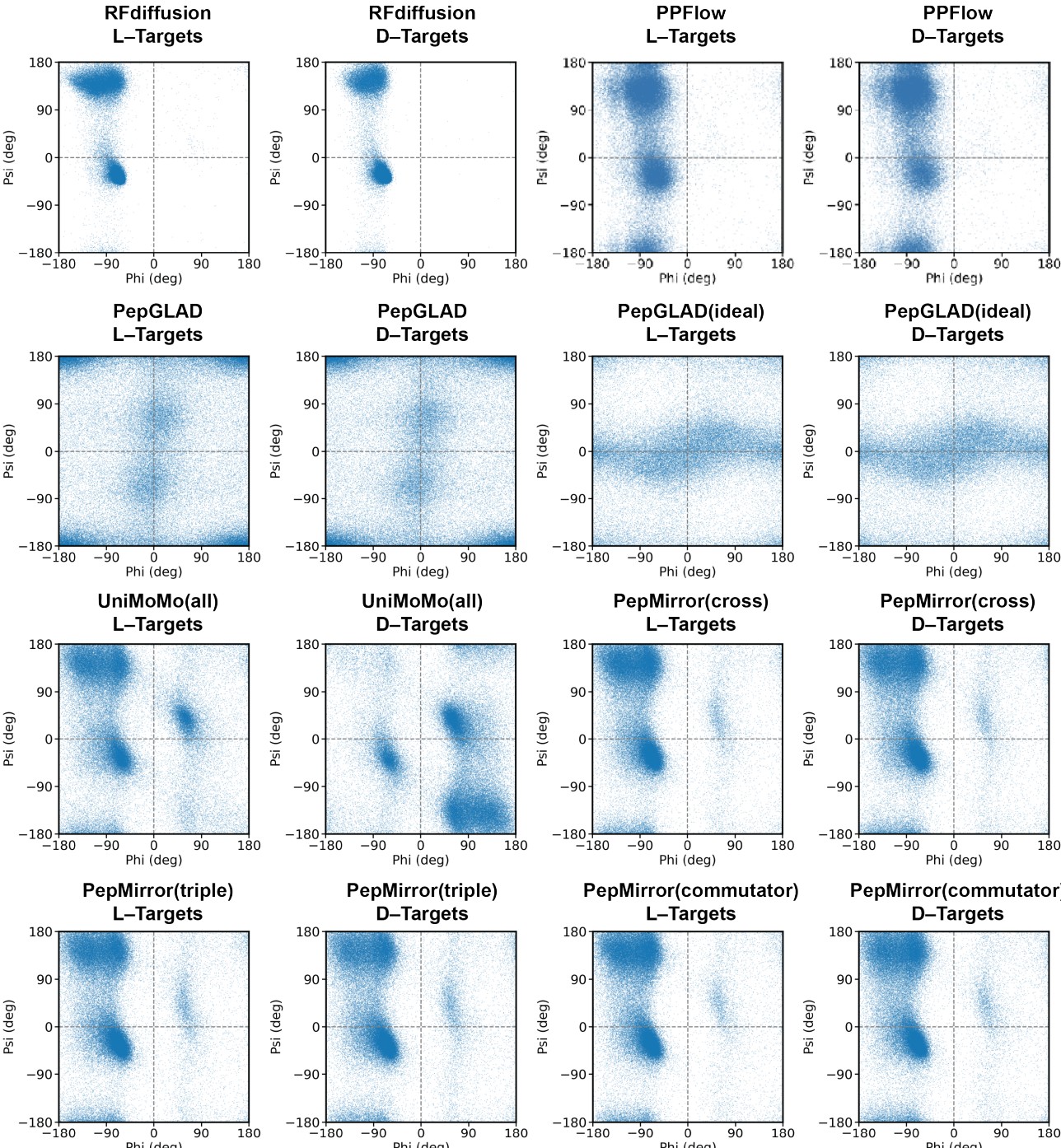

*Figure S7.* Ramachadran plots of generated peptides from certain models.

*Table S7.* Interface quality of of generated L/D peptides evaluated by Rosetta. The best and second best data is labeled orange/light orange for L tasks, and blue/light blue for D tasks.

| Models | Task | Suc. | Avg. | Top | IMP |
|---|---|---|---|---|---|
| RFDiffusion | L | 71.87 | -22.86 | -38.87 | 50.00 |
|  | D | 49.82 | -7.30 | -19.09 | 10.00 |
| DiffPepBuilder | L | 51.82 | -15.37 | -22.58 | 24.44 |
|  | D | 31.30 | -12.66 | -6.66 | 16.67 |
| PepFlow | L | 96.99 | -20.45 | -33.02 | 35.56 |
|  | D | 65.23 | -12.43 | -23.18 | 17.78 |
| D-Flow | L | 96.30 | -18.51 | -30.83 | 27.78 |
|  | D | 72.08 | -12.61 | -24.95 | 17.78 |
| PPFlow | L | 10.15 | -8.93 | -13.14 | 8.89 |
|  | D | 9.1 | -9.05 | -12.17 | 13.33 |
| PepGLAD(ideal) | L | 83.05 | -14.77 | -27.65 | 32.22 |
|  | D | 77.92 | -13.97 | -29.25 | 28.89 |
| PepMirror(cross) | L | 95.98 | -23.27 | -40.66 | 61.11 |
|  | D | 90.88 | -19.90 | -36.07 | 47.78 |
| PepMirror(triple) | L | 96.48 | -23.49 | -40.80 | 58.89 |
|  | D | 91.42 | -20.37 | -36.51 | 45.56 |
| PepMirror(commu.) | L | 97.23 | -24.01 | -41.25 | 60.00 |
|  | D | 91.13 | -20.31 | -36.34 | 42.22 |

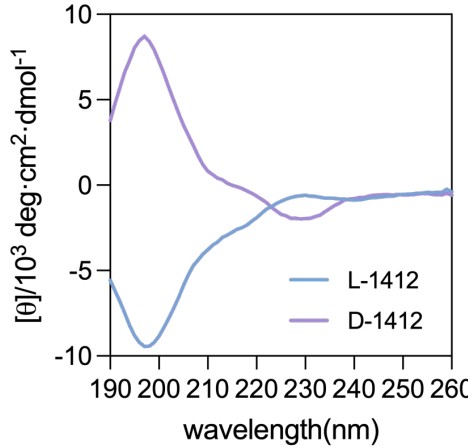

*Figure S8.* Circular dichroism (CD) of D-1412 and its enantiomer L-1412.

## A.7. Extension of AFI

Although we only showcased the application of AFI within the framework of UniMoMo, many variants can be easily designed. For example, the methods to construct axial vectors besides the three listed in this paper, the combination of these axial vectors that may provide complementary informations, the place to inject axial vectors (in FFN, after GNN, or both), and to use pesudo-scalar instead of axial vectors for chirality awareness. Here, we would like to share some preliminary results on testing these variants of AFI, and we believe these indicate the broader potential application of our method.

First, we tested the combination of all three axial vector types (cross, triple product projection, and commutator), where these vectors are all constructed and concatenated with the original polar vector features. The result show that this mixed version does not show much improvement (Table S8). However, it remains an interesting direction to try different combinations of axial features.

Moreover, instead of projecting triple scalar products into vector channels, we tested injecting them directly into node scalar features (denoted as triple_scalar). We also tested a lightweight variant that applies such mixing only once after GNN-based feature initialization (denoted as triple_scalar_once). These variants still achieve similarly performance, while one-time injection leads to a slight drop in chirality consistency (Table S9). These results suggest that AFI does not have to

*Table S8.* The performance of the model that mixed all three types of axial feature as described in the main text

| Task | Right Chirality% | | Vina Score | | | |
|---|---|---|---|---|---|---|
| | Raw | Minimized | Suc.% | Avg. | Top | IMP% |
| L | 99.95 | 99.86 | 99.61 | -4.27 | -5.89 | 72.83 |
| D | 99.97 | 99.87 | 99.70 | -4.18 | -5.75 | 75.00 |

rely on vector channels or EPT. Pseudo-scalar features can be easily constructed from a GNN, which can be injected into architectures without vector channels.

*Table S9.* Performances of the model variants that use pseudo-scalar features

| Variant | Task | Right Chirality% | | Vina Score | | | |
|---|---|---|---|---|---|---|---|
| | | Raw | Minimized | Suc.% | Avg. | Top | IMP% |
| triple_scalar | L | 99.50 | 99.37 | 99.57 | -4.31 | -5.91 | 76.34 |
| | D | 99.33 | 99.19 | 99.61 | -4.22 | -5.88 | 68.82 |
| triple_scalar_once | L | 98.04 | 97.99 | 99.72 | -4.25 | -5.88 | 72.04 |
| | D | 97.05 | 96.98 | 99.70 | -4.17 | -5.76 | 65.59 |

# B. Theory Details

## B.1. Finding axial vector features via decomposition of the tensor product of SO(3) representations

This subsection recaps the minimal $SO(3)$ representation theory we use and uses the concrete Cartesian formulas that yield the dot product, cross product, and the symmetric-traceless tensor $M(u)$. We then explain how our three axial features are obtained by composing these low-order geometric components.

**Basic notation for $SO(3)$ representations.**   Let $SO(3)$ act on $V := \mathbb{R}^3$ by the standard (geometric) action $v \mapsto Rv$. This 3-dimensional representation is the irreducible representation of angular momentum index $l = 1$, commonly denoted $V^{(1)}$ ($\dim V^{(\ell)} = 2\ell + 1, \ell \geq 0$). For any two representations $U, W$ of $SO(3)$, the tensor-product representation $U \otimes W$ is defined by

$$R \cdot (u \otimes w) := (Ru) \otimes (Rw), \qquad \forall R \in SO(3), u \in U, w \in W, \tag{21}$$

extended linearly.

A fundamental problem in representation theory is to decompose the tensor products into irreducible representations. The Clebsch-Gordan decomposition (see (Fulton & Harris, 2004; Hall, 2015), or (Thomas et al., 2018) for a machine learning perspective) is such a result: for irreducible representations $V^{(\ell_1)}$ and $V^{(\ell_2)}$,

$$V^{(\ell_1)} \otimes V^{(\ell_2)} \cong \bigoplus_{J=|\ell_1-\ell_2|}^{\ell_1+\ell_2} V^{(J)}. \tag{22}$$

In particular,

$$V^{(1)} \otimes V^{(1)} \cong V^{(0)} \oplus V^{(1)} \oplus V^{(2)}. \tag{23}$$

**Cartesian realization via rank-2 tensors.**   Identify $V \otimes V$ with the space of rank-2 Cartesian tensors (matrices) by

$$\Phi : V \otimes V \to \mathbb{R}^{3\times 3}, \qquad \Phi(u \otimes v) = u\,v^\top. \tag{24}$$

Under this identification, the $SO(3)$ action becomes conjugation $A \mapsto RAR^\top$. A convenient explicit realization of the three irreducible summands in (23) is given by the classical decomposition of a rank-2 tensor $T_{ij} = u_i v_j$ into (i) scalar representation, (ii) vector representation, and (iii) traceless symmetric tensor representation; see, e.g., the formulas

summarized in (Foster, 2021):

$$
\begin{align}
\text{(scalar, } l = 0) \quad & \mathcal{P}_0(T) := T_{kk} = u \cdot v, \tag{25} \\
\text{(vector, } l = 1) \quad & (\mathcal{P}_1(T))_i := \epsilon_{ijk} T_{jk} = (u \times v)_i, \tag{26} \\
\text{(traceless sym., } l = 2) \quad & (\mathcal{P}_2(T))_{ij} := \frac{1}{2}(T_{ij} + T_{ji}) - \frac{1}{3}\delta_{ij} T_{kk}, \tag{27}
\end{align}
$$

where we employ the Einstein summation convention throughout, and $\epsilon_{ijk}$ is the Levi-Civita symbol, which is antisymmetric in all indices with $\epsilon_{123} = 1$, $\mathcal{P}_i$ is the component-extraction map. Equations (25)–(27) realize the decomposition (23) in a basis-free Cartesian form: $\mathcal{P}_0$ extracts the $l = 0$ component, $\mathcal{P}_1$ extracts the $l = 1$ component, and $\mathcal{P}_2$ extracts the $l = 2$ component.

**The symmetric–traceless tensor $M(u)$.** As another geometric component, specializing (27) to $u = v$ yields the standard traceless symmetric tensor

$$
M(u) := \mathcal{P}_2(u \otimes u) = uu^\top - \frac{1}{3}\|u\|^2 I. \tag{28}
$$

This tensor carries the $l = 2$ irreducible representation $V^{(2)}$.

**From low-order components to our three axial features.** In our model, the input provides multiple $l = 1$ vector channels; denote three such polar vector channels by $u, v, w \in \mathbb{R}^3$. The decomposition above provides two key geometric building blocks: (i) the $l = 1$ coupling $u \times v$ from (26), and (ii) the $l = 2$ object $M(u)$ from (28). We then form three (axial) vector features by simple compositions of these primitives:

$$
\begin{align}
\mathbf{f}_1 &:= u \times v, \tag{29} \\
\mathbf{f}_2 &:= \left((u \times v) \cdot w\right) w, \tag{30} \\
\mathbf{f}_3 &:= \mathrm{ax}\left([M(v), M(u)]\right) = -(u \cdot v)(u \times v), \tag{31}
\end{align}
$$

where $[A, B] = AB - BA$ is the matrix commutator and $\mathrm{ax}(\cdot)$ maps an antisymmetric matrix $A$ to its axial vector $a \in \mathbb{R}^3$ defined by $A_{ij} = \epsilon_{ijk} a_k$.

Feature $\mathbf{f}_1$ captures the oriented normal of the $(u, v)$-plane. $\mathbf{f}_2$ injects signed-volume information via the scalar triple product. $\mathbf{f}_3$ vanishes when $u \perp v$ or $u \parallel v$, encoding a distinct coupling between $u$ and $v$. In practice, for numerical stability in the deep Neural Network, we use normalization in some channels, see Alg. 1 for AFI implementation of the above three axial vector features.

*Remark* B.1. In principle, one can construct infinitely many axial vector features from polar vector channels by composing higher-order tensor products and contractions. In practice, we use only the three low-order features above for their computational efficiency and ease of implementation. Our construction is not intended to enumerate all possible axial features, nor to claim optimality.

### B.2. Equivariance proof

*proof of Proposition 3.5.* It is direct to check the parity of axial features: if we reflect $u, v, w \mapsto -u, -v, -w$, the features will not change sign. Since the feature mixing is channel-wise and the polar vector features are $E(3)$-equivariant, we only need to check the axial features are $SE(3)$-equivariant. Let $g = (R, t) \in SE(3)$ with $R \in SO(3)$. The three features are built from *vector* inputs $u, v$ (and possibly $w$), translations do not act on these vectors since the initialization (embedding) of the input molecules uses the differences of vectors (Jiao et al., 2024; Kong et al., 2025b), we only need to check $SO(3)$-equivariance. We write $g \cdot u := Ru$.

**(1) Cross product.** Define $\mathbf{f}_1(u, v) := u \times v \in \mathbb{R}^3$. For any $R \in SO(3)$ and any $x \in \mathbb{R}^3$,

$$
x \cdot \left((Ru) \times (Rv)\right) = \det[x, Ru, Rv] = \det[R^\top x, u, v] = (R^\top x) \cdot (u \times v) = x \cdot R(u \times v),
$$

where we used the scalar triple product identity $a \cdot (b \times c) = \det[a, b, c]$ and $\det R = 1$. Since this holds for all $x$, we get

$$
(Ru) \times (Rv) = R(u \times v),
$$

**Algorithm 1** Axial Feature Injection (AFI).

*Note:* The `if/else` structure is for illustrative purposes only. In implementation, each TYPE should be compiled into a separate function or module to eliminate runtime branching and ensure efficient execution.

---

**Input:** polar ($E(3)$-equivariant) vector channels $V \in \mathbb{R}^{N \times 3 \times K}$

**Output:** polar-axial mixing (only $SE(3)$-equivariant) vector channels $\widetilde{V} \in \mathbb{R}^{N \times 3 \times K}$

**Parameters:** an unbiased linear layer `Linear` $: \mathbb{R}^{2K} \to \mathbb{R}^{K}$ applied along the channel dimension

**Choice:** axial vector feature TYPE $\in \{$CROSS, TRIPLE, COMMUTATOR$\}$

**Channel shift (wrap-around) and normalization:**

$V^{(1)} \leftarrow \texttt{roll}(V, -1, \dim = -1)$ ⟶ (next channel)

$\widehat{V}^{(1)} \leftarrow \texttt{normalize}(V^{(1)}, \dim = -2, \text{keepdim} = \text{True})$ ⟶ (normalize to unit vector in $\mathbb{R}^3$ )

**Construct axial channels $A \in \mathbb{R}^{N \times 3 \times K}$:**

**if** TYPE $=$ CROSS **then**
  $A \leftarrow \texttt{cross}(V, \widehat{V}^{(1)}, \dim = -2)$

**else if** TYPE $=$ TRIPLE **then**
  $V^{(2)} \leftarrow \texttt{roll}(V, -2, \dim = -1)$ ⟶ (next-next channel)
  $\widehat{V}^{(2)} \leftarrow \texttt{normalize}(V^{(2)}, \dim = -2, \text{keepdim} = \text{True})$
  $C \leftarrow \texttt{cross}(V, \widehat{V}^{(1)}, \dim = -2)$
  $s \leftarrow \texttt{dot}(C, \widehat{V}^{(2)}, \dim = -2, \text{keepdim} = \text{True})$
  $A \leftarrow s \odot \widehat{V}^{(2)}$

**else if** TYPE $=$ COMMUTATOR **then**
  $V_{\text{norm}} \leftarrow \texttt{normalize}(V, \dim = -2, \text{keepdim} = \text{True})$
  $s \leftarrow \texttt{dot}(V_{\text{norm}}, \widehat{V}^{(1)}, \dim = -2, \text{keepdim} = \text{True})$
  $C \leftarrow \texttt{cross}(V, \widehat{V}^{(1)}, \dim = -2)$
  $A \leftarrow s \odot C$

**end if**

**Inject axial information by channel mixing:**

$Z \leftarrow \texttt{Concat}(V, A; \dim = -1) \in \mathbb{R}^{N \times 3 \times 2K}$

$\widetilde{V} \leftarrow \texttt{Linear}(Z)$ ⟶ (applied to the last dimension, no bias)

**return** $\widetilde{V}$

---

i.e. $\mathbf{f}_1$ is $SO(3)$-equivariant (hence $SE(3)$-equivariant).

**(2) Scalar triple product times a vector.** Let $\tau(u, v, w) := w \cdot (u \times v) \in \mathbb{R}$ and define the vector feature

$$\mathbf{f}_2(u, v, w) := \tau(u, v, w)\, w = \big(w \cdot (u \times v)\big)\, w \in \mathbb{R}^3.$$

Using the same determinant identity and $\det R = 1$,

$$\tau(Ru, Rv, Rw) = (Rw) \cdot \big((Ru) \times (Rv)\big) = \det[Rw, Ru, Rv] = \det[w, u, v] = \tau(u, v, w),$$

so $\tau$ is $SO(3)$-*invariant*. Therefore

$$\mathbf{f}_2(Ru, Rv, Rw) = \tau(Ru, Rv, Rw)\,(Rw) = \tau(u, v, w)\, Rw = R\, f_2(u, v, w),$$

so $\mathbf{f}_2$ is $SO(3)$-equivariant (hence $SE(3)$-equivariant).

**(3) Commutator feature via traceless rank-2 tensors.** Let

$$M(u) := uu^\top - \frac{1}{3}\|u\|^2 I \in \mathbb{R}^{3 \times 3}.$$

Define

$$\mathbf{f}_3(u, v) := \text{ax}\big([M(v), M(u)]\big) = \text{ax}\big([vv^\top, uu^\top]\big).$$

For any $x \in \mathbb{R}^3$,

$$[vv^\top, uu^\top]x = v(v^\top u)(u^\top x) - u(u^\top v)(v^\top x) = (u \cdot v)\big(v(u \cdot x) - u(v \cdot x)\big).$$

Using the vector triple product identity $(a \times b) \times x = b(a \cdot x) - a(b \cdot x)$, we get

$$v(u \cdot x) - u(v \cdot x) = (u \times v) \times x,$$

so

$$[vv^\top, uu^\top]x = (u \cdot v)(u \times v) \times x.$$

Therefore $[vv^\top, uu^\top]$ is the cross-product matrix of the axial vector

$$(u \cdot v)(u \times v),$$

which is equivalent up to the convention of the hat map (i.e., $\widehat{\omega}x = \omega \times x$). We obtain $\mathbf{f}_3(u,v) = (u \cdot v)(u \times v)$. Therefore, since $(Ru) \cdot (Rv) = u \cdot v$ and $(Ru) \times (Rv) = R(u \times v)$ (by part (1)),

$$\mathbf{f}_3(Ru, Rv) = ((Ru) \cdot (Rv))\,((Ru) \times (Rv)) = (u \cdot v)\,R(u \times v) = R\,\mathbf{f}_3(u,v),$$

so $\mathbf{f}_3$ is $SO(3)$-equivariant.

Combining the three parts, all listed features are $SE(3)$-equivariant with respect to $(u,v)$ (and $w$ in (2)).

$\square$

### B.3. Chirality awareness with AFI

#### B.3.1. SETUP AND NOTATION.

Fix a sample $X$ and consider a fixed node index $i \in \{1, \ldots, N\}$ and an output channel index $k \in \{1, \ldots, K\}$. Throughout Appendix B.3, $H(X)$ and $V(X)$ refer to the intermediate features before the channel-wise MLP $\varphi$. In particular, we write

$$V_i(X) = \big(v_{i,1}(X), \ldots, v_{i,K}(X)\big), \qquad v_{i,k}(X) \in \mathbb{R}^3,$$

for the polar vector channels at this stage (i.e., *before* the feature mixing at the end of this block).

Based on the polar vector features, we also construct axial vector feature channels like 3.3,

$$A_i(X) = \big(a_{i,1}(X), \ldots, a_{i,K}(X)\big), \qquad a_{i,k}(X) \in \mathbb{R}^3,$$

computed from the same post-attention, pre-FFN representations.

By definition, polar vectors change sign while axial vectors do not:

$$v_{i,k}(-X) = -v_{i,k}(X), \qquad a_{i,k}(-X) = a_{i,k}(X), \qquad k = 1, \ldots, K. \tag{32}$$

#### B.3.2. LINEAR FEATURE MIXING

Let $A_k, B_k \in \mathbb{R}^K$ be channel-wise mixing coefficients (corresponding to the channel-mixing linear map applied after concatenating $(v, a)$).

To quantify the typical discrepancy behavior, we introduce a mild distributional assumption on the mixing coefficients $(A_k, B_k)$. Our arguments only require that the coordinates are *independent sub-Gaussian* for concentration. For notational convenience and to keep constants explicit, we state the assumption in the Gaussian case. The same proof extends verbatim to any independent sub-Gaussian distributions with comparable parameters (For example, bounded valued distributions).

**Assumption B.2** (Mixing coefficients distribution)**.** For every $k = 1, \ldots, K$, $A_k$ and $B_k$ are independent Gaussian vectors

$$A_k \sim \mathcal{N}(0, \sigma_A^2 I_K), \qquad B_k \sim \mathcal{N}(0, \sigma_B^2 I_K),$$

for some $\sigma_A, \sigma_B > 0$.

Define the mixed vector features

$$\widetilde{v}_{i,k}(X) := A_k^\top v_{i,:}(X) + B_k^\top a_{i,:}(X) \in \mathbb{R}^3, \tag{33}$$

where $v_{i,:}(X) := (v_{i,1}(X), \ldots, v_{i,K}(X))$ and $A_k^\top v_{i,:}(X)$ means the linear combination

$$A_k^\top v_{i,:}(X) = \sum_{\ell=1}^K (A_k)_\ell \, v_{i,\ell}(X) \in \mathbb{R}^3,$$

and similarly $B_k^\top a_{i,:}(X) = \sum_{\ell=1}^K (B_k)_\ell \, a_{i,\ell}(X) \in \mathbb{R}^3$.

Note that the scalar part of the latent code is $c(X) = \varphi(H(X), \|\widetilde{V}(X)\|)$. Under the central reflection,

$$\varphi(-X) = \varphi([H(X), \|A^\top v(X) + B^\top a(X)\|]) \tag{34}$$

$$\varphi(X) = \varphi([H(X), \| - A^\top v(X) + B^\top a(X)\|]). \tag{35}$$

The key quantity is the parity-induced norm difference

$$\Delta_{i,k}(X) := \big| \, \|\widetilde{v}_{i,k}(X)\| - \|\widetilde{v}_{i,k}(-X)\| \, \big|.$$

**Assumption B.3** (Boundedness of vector features). There exist constants $S_v, S_a > 0$ such that for every fixed sample $X$ and node $i$,

$$\|v_{i,:}(X)\|_F^2 := \sum_{\ell=1}^K \|v_{i,\ell}(X)\|^2 \le S_v^2, \qquad \|a_{i,:}(X)\|_F^2 := \sum_{\ell=1}^K \|a_{i,\ell}(X)\|^2 \le S_a^2.$$

**Assumption B.4** (Non-degenerate polar–axial correlation). Define the correlation matrix $C = C_i(X) \in \mathbb{R}^{K \times K}$ by

$$C_{k\ell} := \langle v_{i,k}(X), a_{i,\ell}(X) \rangle \qquad (1 \le k, \ell \le K). \tag{36}$$

Assume

$$\|C\|_F \ge \tau > 0 \quad \text{and} \quad r_{\mathrm{eff}}(C) := \frac{\|C\|_F^2}{\|C\|_{\mathrm{op}}^2} \ge r_0,$$

for some constants $\tau > 0$ and $r_0 \ge 1$. Note that $\|C\|_F^2$ is the square sum of all singular values of $C$, $\|C\|_{\mathrm{op}}$ is the maximal singular value of $C$. $r_{\mathrm{eff}}(C)$ is the so-called efficient rank, a quantitative value of the rank of matrix $C$.

**Lemma B.5** (Parity flips only the polar part). *Let*

$$p := A_k^\top v_{i,:}(X) \in \mathbb{R}^3, \qquad q := B_k^\top a_{i,:}(X) \in \mathbb{R}^3.$$

*Then*

$$\widetilde{v}_{i,k}(X) = p + q, \qquad \widetilde{v}_{i,k}(-X) = -p + q.$$

*Consequently,*

$$\|\widetilde{v}_{i,k}(X)\|^2 - \|\widetilde{v}_{i,k}(-X)\|^2 = \|p + q\|^2 - \| - p + q\|^2 = 4\langle p, q \rangle, \tag{37}$$

*and*

$$\Delta_{i,k}(X) = \frac{4|\langle p, q \rangle|}{\|p + q\| + \|q - p\|} \ge \frac{2|\langle p, q \rangle|}{\|p\| + \|q\|}. \tag{38}$$

*Proof.* The parity rule (32) implies

$$A_k^\top v_{i,:}(-X) = \sum_{\ell=1}^K (A_k)_\ell \, v_{i,\ell}(-X) = \sum_{\ell=1}^K (A_k)_\ell \, (-v_{i,\ell}(X)) = -p,$$

while

$$B_k^\top a_{i,:}(-X) = \sum_{\ell=1}^K (B_k)_\ell \, a_{i,\ell}(-X) = \sum_{\ell=1}^K (B_k)_\ell \, a_{i,\ell}(X) = q.$$

Thus $\widetilde{v}_{i,k}(-X) = -p + q$ and $\widetilde{v}_{i,k}(X) = p + q$.

To prove (37), expand both squares:

$$\|p + q\|^2 = \|p\|^2 + \|q\|^2 + 2\langle p, q \rangle, \qquad \| - p + q\|^2 = \|p\|^2 + \|q\|^2 - 2\langle p, q \rangle.$$

Subtracting gives $\|p + q\|^2 - \| - p + q\|^2 = 4\langle p, q\rangle$.

For (38), use the identity $|a - b| = \frac{|a^2 - b^2|}{a + b}$ for $a, b \geq 0$:

$$\Delta_{i,k}(X) = \big| \|p + q\| - \|q - p\| \big| = \frac{\big| \|p + q\|^2 - \|q - p\|^2 \big|}{\|p + q\| + \|q - p\|} = \frac{4|\langle p, q\rangle|}{\|p + q\| + \|q - p\|}.$$

Finally, by triangle inequality,

$$\|p + q\| \leq \|p\| + \|q\|, \qquad \|q - p\| \leq \|p\| + \|q\|,$$

hence $\|p + q\| + \|q - p\| \leq 2(\|p\| + \|q\|)$, which yields

$$\Delta_{i,k}(X) \geq \frac{4|\langle p, q\rangle|}{2(\|p\| + \|q\|)} = \frac{2|\langle p, q\rangle|}{\|p\| + \|q\|}.$$

$\square$

**Lemma B.6** (Small-ball bound for a Gaussian bilinear form). *Let $A \sim \mathcal{N}(0, \sigma_A^2 I_K)$ and $B \sim \mathcal{N}(0, \sigma_B^2 I_K)$ be independent, and let $C \in \mathbb{R}^{K \times K}$ be fixed. Define the bilinear form*

$$S := A^\top C B. \tag{39}$$

*Then for any $\varepsilon \in (0, 1)$,*

$$\mathbb{P}\Big( |S| \leq \varepsilon \, \sigma_A \sigma_B \, \|C\|_F \Big) \leq \frac{2}{\sqrt{\pi}} \varepsilon + 2 \exp\big( - c \, r_{\text{eff}}(C) \big), \tag{40}$$

*where $c > 0$ is an absolute constant and $r_{\text{eff}}(C) = \|C\|_F^2 / \|C\|_{\text{op}}^2$.*

*Proof.* **Step 1 (Condition on $A$).** Since $B$ is Gaussian and independent of $A$, then conditioned on $A$,

$$S \mid A = A^\top C B \sim \mathcal{N}\big(0, \, \sigma_B^2 \, \|C^\top A\|^2\big).$$

Let $Z \sim \mathcal{N}(0, 1)$. Then $S \mid A \sim \sigma_B \|C^\top A\| Z$ in distribution. Hence for any $t > 0$,

$$\mathbb{P}\big(|S| \leq t \mid A\big) = \mathbb{P}\Big(|Z| \leq \frac{t}{\sigma_B \|C^\top A\|}\Big) \leq \sqrt{\frac{2}{\pi}} \frac{t}{\sigma_B \|C^\top A\|}, \tag{41}$$

where we used the standard bound for $Z \sim \mathcal{N}(0, 1)$: $\mathbb{P}(|Z| \leq u) \leq \sqrt{\frac{2}{\pi}} u$ for $u \geq 0$.

**Step 2 (Lower bound $\|C^\top A\|$ via Hanson-Wright).** Write $A = \sigma_A g$ with $g \sim \mathcal{N}(0, I_K)$, and set

$$M := CC^\top \succeq 0.$$

Then

$$\|C^\top A\|^2 = \sigma_A^2 \|C^\top g\|^2 = \sigma_A^2 \, g^\top (CC^\top) \, g = \sigma_A^2 \, g^\top M g. \tag{42}$$

We apply the Hanson-Wright inequality in the Gaussian case (see (Vershynin, 2018) Theorem 6.2.1). In the notation of that theorem:

$$X = g \in \mathbb{R}^K, \quad \text{(mean-zero, independent, sub-Gaussian coordinates with } \|X_i\|_{\psi_2} \lesssim 1), \quad A_{\text{HW}} = M.$$

The theorem states that there exist absolute constants $c, C > 0$ such that for all $t \geq 0$,

$$\mathbb{P}\Big( |g^\top M g - \text{tr}(M)| \geq t \Big) \leq 2 \exp\Big( - c \min\Big\{ \frac{t^2}{\|M\|_F^2}, \frac{t}{\|M\|_{\text{op}}} \Big\} \Big). \tag{43}$$

Here

$$\text{tr}(M) = \text{tr}(CC^\top) = \|C\|_F^2, \qquad \|M\|_F = \|CC^\top\|_F, \qquad \|M\|_{\text{op}} = \|CC^\top\|_{\text{op}} = \|C\|_{\text{op}}^2.$$

We take $t = \frac{1}{2}\mathrm{tr}(M) = \frac{1}{2}\|C\|_F^2$ in (43). Then

$$\mathbb{P}\Big(g^\top Mg \leq \tfrac{1}{2}\mathrm{tr}(M)\Big) \leq \mathbb{P}\Big(|g^\top Mg - \mathrm{tr}(M)| \geq \tfrac{1}{2}\mathrm{tr}(M)\Big) \leq 2\exp\Big(-c\min\Big\{\frac{\mathrm{tr}(M)^2}{4\|M\|_F^2}, \frac{\mathrm{tr}(M)}{2\|M\|_{\mathrm{op}}}\Big\}\Big).$$

Using $\|M\|_F^2 \leq \|M\|_{\mathrm{op}}\mathrm{tr}(M)$ (since $M \succeq 0$), we get

$$\frac{\mathrm{tr}(M)^2}{\|M\|_F^2} \geq \frac{\mathrm{tr}(M)}{\|M\|_{\mathrm{op}}}.$$

Hence the minimum is controlled by the second term, and we obtain

$$\mathbb{P}\Big(g^\top Mg \leq \tfrac{1}{2}\|C\|_F^2\Big) \leq 2\exp\Big(-c'\frac{\|C\|_F^2}{\|C\|_{\mathrm{op}}^2}\Big) = 2\exp\big(-c'\, r_{\mathrm{eff}}(C)\big), \tag{44}$$

where $c' = c/2 > 0$ is an absolute constant and $r_{\mathrm{eff}}(C) := \|C\|_F^2/\|C\|_{\mathrm{op}}^2$ is the effective rank. Therefore, defining the event

$$\mathcal{E} := \Big\{\|C^\top A\| \geq \sigma_A\|C\|_F/\sqrt{2}\Big\},$$

we have from (42) and (44) that

$$\mathbb{P}(\mathcal{E}^c) \leq 2\exp\big(-c'\, r_{\mathrm{eff}}(C)\big). \tag{45}$$

**Step 3 (Standard conditional small-ball bound).** Set $t := \varepsilon\sigma_A\sigma_B\|C\|_F$. On $\mathcal{E}$ we have $\|C^\top A\| \geq \sigma_A\|C\|_F/\sqrt{2}$, thus by (41),

$$\mathbb{P}\big(|S| \leq t \mid A\big) \leq \sqrt{\frac{2}{\pi}}\,\frac{\varepsilon\sigma_A\sigma_B\|C\|_F}{\sigma_B(\sigma_A\|C\|_F/\sqrt{2})} = \frac{2}{\sqrt{\pi}}\varepsilon.$$

Therefore,

$$\mathbb{P}(|S| \leq t) \leq \mathbb{P}(|S| \leq t, \mathcal{E}) + \mathbb{P}(\mathcal{E}^c) \leq \frac{2}{\sqrt{\pi}}\varepsilon + 2\exp\big(-c\, r_{\mathrm{eff}}(C)\big).$$

This is exactly (40). $\qquad\square$

**Lemma B.7** (Gaussian norm bound). *If $G \sim \mathcal{N}(0, I_K)$, then for any $t \geq 0$,*

$$\mathbb{P}\big(\|G\| \geq \sqrt{K} + t\big) \leq e^{-t^2/2}.$$

*In particular,*

$$\mathbb{P}\big(\|G\| \leq 2\sqrt{K}\big) \geq 1 - e^{-K/2}.$$

*Proof.* This is a standard Gaussian concentration bound for the Lipschitz function $g \mapsto \|g\|$ with Lipschitz constant 1. For completeness: by Gaussian isoperimetry (or concentration of measure), $\mathbb{P}(\|G\| \geq \mathbb{E}\|G\| + t) \leq e^{-t^2/2}$. Since $\mathbb{E}\|G\| \leq \sqrt{K}$, we get the displayed inequality. Setting $t = \sqrt{K}$ yields $\mathbb{P}(\|G\| \leq 2\sqrt{K}) \geq 1 - e^{-K/2}$. $\qquad\square$

**Theorem B.8** (Chirality awareness, discrepancy). *Fix a sample $X$, node $i$ and channel $k$. Assume (33) and the parity rule (32). Under Assumptions B.3–B.4, define $C$ by (36). Let $\Delta_{i,k}(X) = \big|\|\widetilde{v}_{i,k}(X)\| - \|\widetilde{v}_{i,k}(-X)\|\big|$.*

*Then for any $\varepsilon \in (0, 1)$, with probability at least*

$$1 - \Big(\frac{2}{\sqrt{\pi}}\varepsilon + 2e^{-cr_0} + 2e^{-K/2}\Big),$$

*we have the explicit lower bound*

$$\Delta_{i,k}(X) \geq \frac{c_0\,\varepsilon\,\sigma_A\sigma_B\,\tau}{\sigma_A S_v + \sigma_B S_a}, \tag{46}$$

*where $c_0 > 0$ is an absolute constant (one may take $c_0 = \frac{1}{2}$ after absorbing constant factors). In particular, $\Delta_{i,k}(X)$ is bounded away from 0 with high probability.*

*Moreover, in the absence of AFI (i.e., $B_k \equiv 0$ so that $q = 0$), we have $\Delta_{i,k}(X) = 0$ deterministically.*

*In the informal version Theorem 3.1, the two constants are*

$$\delta_W(\varepsilon) = \frac{2}{\sqrt{\pi}}\varepsilon + 2e^{-cr_0} + 2e^{-K/2}, \qquad c_W = \frac{c_0\,\sigma_A\sigma_B\,\tau}{\sigma_A S_v + \sigma_B S_a}.$$

*Proof.* **Step 1 (Reduce norm separation to an inner product).** By Lemma B.5, for $p = A_k^\top v_{i,:}(X)$ and $q = B_k^\top a_{i,:}(X)$,

$$\Delta_{i,k}(X) \;\geq\; \frac{2|\langle p, q \rangle|}{\|p\| + \|q\|}. \tag{47}$$

**Step 2 (Express $\langle p, q \rangle$ as a bilinear form).** Using linearity and the definition of $C$ in (36),

$$\langle p, q \rangle = \Big\langle \sum_{\ell=1}^K (A_k)_\ell v_{i,\ell}(X), \; \sum_{m=1}^K (B_k)_m a_{i,m}(X) \Big\rangle = \sum_{\ell=1}^K \sum_{m=1}^K (A_k)_\ell (B_k)_m \langle v_{i,\ell}(X), a_{i,m}(X) \rangle = A_k^\top C B_k.$$

Therefore, if we set $S := A_k^\top C B_k$,

$$|\langle p, q \rangle| = |S|. \tag{48}$$

**Step 3 (Lower bound $|S|$ with high probability).** Apply Lemma B.6 to $S = A_k^\top C B_k$. Since $r_{\text{eff}}(C) \geq r_0$ by Assumption B.4, for any $\varepsilon \in (0, 1)$, with probability at least

$$1 - \Big( \frac{2}{\sqrt{\pi}} \varepsilon + 2e^{-cr_0} \Big), \tag{49}$$

we have

$$|S| \;\geq\; \varepsilon \, \sigma_A \sigma_B \, \|C\|_F \;\geq\; \varepsilon \, \sigma_A \sigma_B \, \tau, \tag{50}$$

where the second inequality uses $\|C\|_F \geq \tau$.

**Step 4 (Upper bound the denominator $\|p\| + \|q\|$ with high probability).** Write $A_k = \sigma_A G_A$ and $B_k = \sigma_B G_B$ with $G_A, G_B \sim \mathcal{N}(0, I_K)$ independent. First, by Cauchy–Schwarz,

$$\|p\| = \Big\| \sum_{\ell=1}^K (A_k)_\ell v_{i,\ell}(X) \Big\| \leq \sum_{\ell=1}^K |(A_k)_\ell| \, \|v_{i,\ell}(X)\| \leq \Big( \sum_{\ell=1}^K (A_k)_\ell^2 \Big)^{1/2} \Big( \sum_{\ell=1}^K \|v_{i,\ell}(X)\|^2 \Big)^{1/2} = \|A_k\| \, \|v_{i,:}(X)\|_F.$$

Similarly,

$$\|q\| \leq \|B_k\| \, \|a_{i,:}(X)\|_F.$$

Hence,

$$\|p\| + \|q\| \;\leq\; \|A_k\| \, \|v_{i,:}(X)\|_F + \|B_k\| \, \|a_{i,:}(X)\|_F. \tag{51}$$

Now use Assumption B.3 to bound $\|v_{i,:}(X)\|_F \leq S_v$ and $\|a_{i,:}(X)\|_F \leq S_a$:

$$\|p\| + \|q\| \leq \|A_k\| \, S_v + \|B_k\| \, S_a.$$

Next, apply Lemma B.7 to $G_A$ and $G_B$: with probability at least $1 - 2e^{-K/2}$,

$$\|G_A\| \leq 2\sqrt{K} \quad \text{and} \quad \|G_B\| \leq 2\sqrt{K}.$$

On this event,

$$\|A_k\| = \sigma_A \|G_A\| \leq 2\sigma_A \sqrt{K}, \qquad \|B_k\| = \sigma_B \|G_B\| \leq 2\sigma_B \sqrt{K},$$

and thus

$$\|p\| + \|q\| \leq 2\sqrt{K} \, (\sigma_A S_v + \sigma_B S_a). \tag{52}$$

(If one prefers to remove the factor $\sqrt{K}$, one may instead normalize $A_k, B_k$ at initialization; we keep the explicit dependence here.)

**Step 5 (Combine bounds).** Intersect the event (50) and the event (52). By a union bound, this intersection holds with probability at least

$$1 - \Big( \frac{2}{\sqrt{\pi}} \varepsilon + 2e^{-cr_0} + 2e^{-K/2} \Big).$$

On this intersection, substitute (50) into (47) and then use (52):

$$\Delta_{i,k}(X) \geq \frac{2|S|}{\|p\| + \|q\|} \geq \frac{2 \cdot \varepsilon \sigma_A \sigma_B \tau}{2\sqrt{K}(\sigma_A S_v + \sigma_B S_a)} = \frac{\varepsilon \sigma_A \sigma_B \tau}{\sqrt{K}(\sigma_A S_v + \sigma_B S_a)}.$$

This proves (46) up to an absolute constant $c_0$ (absorbing $\sqrt{K}$ if one uses the normalized convention for $A_k, B_k$; otherwise keep the explicit $\sqrt{K}$ factor as above).

**Step 6 (No AFI implies no separation).** If $B_k \equiv 0$, then $q \equiv 0$ and $\widetilde{v}_{i,k}(X) = p$, $\widetilde{v}_{i,k}(-X) = -p$. Thus $\|\widetilde{v}_{i,k}(X)\| = \|\widetilde{v}_{i,k}(-X)\|$ and $\Delta_{i,k}(X) = 0$ deterministically. $\square$

**Corollary B.9** (Chirality awareness, formal). *Assume the latter 2-layer MLP is $\varphi$ that takes the concatenation*

$$\text{scaler}(X) = [\, H(X), \; \|\widetilde{v}_{i,k}(X)\| \,],$$

*as input where $H(X)$ is reflection-invariant (depends only on types, distances, dot products, etc.). Assume $\varphi$ is $\mu$-coercivity in its second argument, i.e.*

$$\|\varphi\big(\text{scaler}(X)\big) - \varphi\big(\text{scaler}(-X)\big)\| \; \geq \; \mu \cdot \Delta_{i,k}(X),$$

*and in particular Theorem B.8 yields a high-probability lower bound on the scalar-output discrepancy. The two constants in 3.1 are*

$$\delta_W = \left( \frac{2}{\sqrt{\pi}}\varepsilon + 2e^{-cr_0} + 2e^{-K/2} \right), \qquad c_W = \frac{c_0\,\mu\,\sigma_A\sigma_B\,\tau}{\sigma_A S_v + \sigma_B S_a}.$$

*Proof.* Since $H(-X) = H(X)$ by parity invariance, the only change in the input to $\varphi$ comes from the norm term. Applying the coercivity property in that coordinate gives the corollary. $\square$

*Remark* B.10. The coercivity assumption is a non-degenerate assumption on $\varphi$. It holds for the general Network parameters. One can analyze its derivative and form concentration arguments as above to show that this non-degeneracy is generic.

### B.4. Discussion on inversion and other orthogonal transforms

**Lemma B.11.** *Let $F \in O(3) \setminus SO(3)$ be any improper orthogonal transform, including a mirror reflection. Set $P = -I_3$ and $Q = FP \in SO(3)$, so that $F = QP$. Assume the polar and axial channels satisfy $SO(3)$-equivariance and the inversion parity rule used in Appendix B.3. Then the discrepancy induced by $F$ is identical to the discrepancy induced by spatial inversion $P$. Consequently, Theorem B.8 extends verbatim from $P = -I_3$ to any $F \in O(3) \setminus SO(3)$.*

*Proof.* Since $\det(F) = -1$ and $\det(P) = -1$, we have $Q = FP \in SO(3)$ and $F = QP$. By $SO(3)$-equivariance and the inversion parity rule,

$$v(FX) = v(QPX) = Qv(PX) = -Qv(X), \qquad a(FX) = a(QPX) = Qa(PX) = Qa(X). \tag{53}$$

Thus, for the AFI mixed feature $\widetilde{v} = A^\top v + B^\top a$, writing $p = A^\top v(X)$ and $q = B^\top a(X)$, we get

$$\widetilde{v}(FX) = Q(-p + q), \qquad \|\widetilde{v}(FX)\| = \| - p + q\|, \tag{54}$$

because $Q$ is orthogonal. On the other hand, spatial inversion gives $\widetilde{v}(PX) = -p + q$. Therefore,

$$\big| \|\widetilde{v}(X)\| - \|\widetilde{v}(FX)\| \big| = \big| \|p + q\| - \| - p + q\| \big| = \big| \|\widetilde{v}(X)\| - \|\widetilde{v}(PX)\| \big|. \tag{55}$$

Hence the improper-transform discrepancy is exactly the same as the inversion discrepancy, so the lower bound in Theorem B.8 applies unchanged. $\square$

## B.5. Proof of diffusion stability

*Proof of Theorem 3.4.* Let $Z_t, Z'_t$ solve (16) with the same Brownian motion and the same initial point $Z_T = Z'_T$ with different conditions $c, c'$.

$$dZ_t = b_\theta(Z_t, t, c)\, dt + \sigma(t)\, dW_t, \tag{56}$$
$$dZ'_t = b_\theta(Z'_t, t, c')\, dt + \sigma(t)\, dW_t. \tag{57}$$

Set $\Delta_t := Z_t - Z'_t$. Subtraction cancels the noise under coupling, giving

$$d\Delta_t = \big(b_\theta(Z_t, t, c) - b_\theta(Z'_t, t, c')\big)\, dt.$$

Note that for the Euclidean norm, we have $\frac{d}{dt}\|x\| \le \|\frac{d}{dt}x\|$ by Cauchy Schwarz inequality. Using Assumption 3.3,

$$\frac{d}{dt}\|\Delta_t\| \le \left\|\frac{d}{dt}\Delta_t\right\| \le \|b_\theta(Z_t, t, c) - b_\theta(Z'_t, t, c')\| \le L_z\|\Delta_t\| + L_c\|c - c'\|. \tag{58}$$

Since $Z_T = Z'_T$, we have $\Delta_T = 0$. Applying Grönwall inequality yields

$$\|\Delta_0\| \le L_c\|c - c'\| \int_0^T e^{L_z s}\, ds = \frac{L_c}{L_z}(e^{L_z T} - 1)\|c - c'\| := K_{\text{diff}}\|c - c'\|.$$

This provides an explicit coupling $(Z_0, Z'_0)$ of $(\mu_c, \mu_{c'})$ such that $\|Z_0 - Z'_0\| = \|\Delta_0\| \le K_{\text{diff}}\|c - c'\|$ holds almost surely. By the definition of the Wasserstein metric,

$$W_2(\mu_c, \mu_{c'}) = \inf_{\Gamma(Z_0, Z'_0)} \sqrt{\mathbb{E}\|Z_0 - Z'_0\|^2} \tag{59}$$

$$= \inf_{\Gamma(Z_0, Z'_0)} \sqrt{\mathbb{E}\|\Delta_0\|^2} \le \frac{L_c}{L_z}(e^{L_z T} - 1)\|c - c'\| \tag{60}$$

where we use a pointwise estimation of the above expectation. We can take $K_{\text{diff}} = \frac{L_c}{L_z}(e^{L_z T} - 1) > 0$. □

## B.6. Initialization: graph embedding

We expand the graph embedding layer of vector features as (Jiao et al., 2024; Kong et al., 2025b). For each node $i$ and channel $k \in \{1, \ldots, K\}$, define the edge vector feature

$$Y_{ij,k}(X) := s_{ij,k}(X)\,(x_i - x_j) \in \mathbb{R}^3, \tag{61}$$

where $s_{ij,k}(X) \in \mathbb{R}$ is a scalar weight depending on the invariant feature (such as atom types) and the RBF of the distance information, and $x_i$ are the 3D coordinates of atom $i$. We assume the $m$ nearest neighbors of atom $i$ is $\mathcal{N}(i)$. Then the initial vector feature after embedding of node $i$ and channel $k$ is a *polar vector feature*

$$v_{i,k}(X) := \frac{1}{m} \sum_{j \in \mathcal{N}(i)} Y_{ij,k}(X) \in \mathbb{R}^3. \tag{62}$$

# C. Experiment Details

## C.1. Shape similarity between residue types

To quantify shape similarity between amino acid pairs, we first generate 1,000 conformations per residue type using RDKit and minimize them with the MMFF94s force field. The resulting conformers are clustered with an RMSD threshold of 0.25 Å, and one representative from each cluster is retained to remove redundancy. For residue types $i$ and $j$, we compute the pairwise shape Tanimoto similarity between every conformer of $i$ and every conformer of $j$, and apply max pooling to obtain a single similarity score for the pair. Concretely, each conformer pair is first aligned by RDKit O3A with shape-based scoring, after which we compute $\mathrm{ShapeTanimotoDist}$; the similarity is defined as $1 - \mathrm{ShapeTanimotoDist}$.

*Table S10.* Hyperparameters and settings for training PepMirror and its variants.

| Models | UniMoMo(pep.) | PepMirror(cross) | PepMirror(triple) | PepMirror(commutator) |
|---|---|---|---|---|
| Optimizer | AdamW | AdamW | AdamW | AdamW |
| $\beta_1$ | 0.9 | 0.9 | 0.9 | 0.9 |
| $\beta_2$ | 0.999 | 0.999 | 0.999 | 0.999 |
| LR | 1e-4 | 1e-4 | 1e-4 | 1e-4 |
| GPU type | A800 | A800 | A800 | A800 |
| Number of GPUs | 8 | 8 | 8 | 8 |
| Days to train (VAE+LDM) | 2 | 2 | 2 | 2 |
| Training epochs (VAE) | 199 | 169 | 179 | 169 |
| Training epochs (LDM) | 375 | 307 | 483 | 448 |

## C.2. Test-Set Preprocessing and Pocket Definition

Because the LNR dataset is curated from the PDB, some entries contain artifacts that can confound parsing and ligand-length determination, including terminal modifications (e.g., N-terminal acetylation and C-terminal amidation), alternative conformations encoded as multiple occupancies (e.g., AGLU/BGLU), and non-protein components such as small molecules, salts, and solvent. To standardize the inputs, we cleaned all structures using Rosetta's `clean_pdb.py` and performed manual inspection to ensure structural completeness and consistency. The resulting curated set, `LNR_clean`, is used for all downstream evaluations.

Binding pockets within the receptors were defined using a CB distance threshold of <10 Å. For Glycine residues, which lack a natural CB atom, a virtual CB was constructed following previously reported protocols. Since these virtual CB atoms are configured for L-amino acids, we synchronized the pocket residue IDs with those identified in the mirror-image complexes to ensure consistency. To ensure a fair comparison, the same pocket inputs were provided to all evaluated models, unless stated otherwise.

## C.3. Training

We largely follow the training protocol of UniMoMo. Specifically, we use the same datasets for linear peptide design (PepBench and ProtFrag) and adopt an identical train/validation split as in UniMoMo. We apply the same two-stage pipeline—training a variational autoencoder (VAE) followed by a latent diffusion model (LDM)—and select the checkpoint with the lowest validation loss for downstream inference and evaluation. All experiments with different axial-vector choices share the same data split and training setup. Detailed hyperparameters are reported in Table S10.

## C.4. Inference

Generally, we generated 100 samples for each targets in the LNR test set with the same length as native peptide binders in cleaned complexes. The random seed was set to 12 for reproducing results. After generation, we reconstructed complexes with full-length receptors for downstream analysis. Some model-specific settings are listed below:

**RFDiffusion**. Among published checkpoints, we employed the `Complex_base_ckpt.pt` for binder design. We found that providing only pocket information leads to pronounced clashes during reconstruction. Therefore, we used full-length structures as input and specify binding sites via hotspots. Following the rule in the training process, we randomly selected 20% pocket residues as hotspot residues for binding site specification, and the same hotspots were used for L and D design. After generating peptide backbones, we sample 1 sequence using ProteinMPNN with `v_48_020.pt` (48 edges with 0.20 Å noise). Following the procedure described in BindCraft, we set the sampling temperature to 0.0001 to make sampling nearly deterministic, yielding the top-ranked sequence according to the model. After integrating the designed sequence into the ligand, we manually added C$\beta$ atoms for all non–Gly residues to enforce the intended initial chirality. We then kept the receptor side-chain conformations fixed to those of the native receptor, and used PDBFixer and OpenMM to sample and optimize the ligand side-chain conformations. The resulting complexes were subsequently subjected to downstream evaluation.

**DiffPepBuilder**. Because this model relies on ESM embeddings as sequence representations, we provide full-length structures as input to ensure that the ESM encoding is well-defined and contextually consistent.

**PPFlow**. We noticed that many designed ligands show severe backbone clashes with receptors, and there is a redock legacy in the released pipeline that heuristically aligns the generated ligand to the native ligand by matching their centroids. However, neither enabling this redocking step nor providing full-length receptor structures could reduce the clash rate. We therefore followed the default setting in the original implementation, using pocket-only inputs without redocking.

**PocketXMol**. The model generates structures containing non-standard amino acids since it does not group atoms into course-grained tokens. For fair comparison, we only evaluated peptides composed of only canonical residues out of 9,300 generated molecules.

### C.5. Implementation of Metrics

**Chirality**. We calculate the chirality of residues based on the scalar triple product: $T = (\vec{N} - \vec{C}_A) \times (\vec{C} - \vec{C}_A) \cdot (\vec{C}_B - \vec{C}_A)$, where a positive $T$ indicates L and a negative $T$ indicates D. For each tested model, we report the proportion of residues that show desired chirality out of all residues, where glycines that are achiral are excluded. As mentioned below, generated structures were minimized before evaluation, and the desired chirality proportion before and after the minimization are both reported.

**Minimization**. We minimize the generated complexes using the Amber ff14SB force field. To preserve the generated geometry, we apply harmonic positional restraints to both the receptor and the ligand during minimization, preventing large deviations from the initial model and ensuring that the minimized structures remain representative for evaluation.

**Interface Energy**. We compute interface energies using AutoDock Vina in score-only mode. Following the UniMoMo definition of interface energy improvement (IMP) (Kong et al., 2025b), we report the proportion of targets for which at least one generated ligand achieves a lower vina score than the reference ligand.

**Binding Surface Recovery**. We define the binding surface as the set of receptor residues whose $C\alpha$ atoms are within $10\,\text{Å}$ of any ligand $C\alpha$ atom. Using the ground-truth complex as reference, we compute for each generated complex the recovery ratio as the fraction of native binding-surface residues that are also present in the generated binding surface.

**Diversity**. Generally, we define diversity as the number of clusters normalized by the total number of samples. For *sequence diversity*, we cluster the 100 generated sequences for each target independently using complete-linkage hierarchical clustering. For *structure diversity*, we first align complexes by the receptor, then compute pairwise ligand $C_\alpha$ RMSDs, and perform complete-linkage clustering with a $2\,\text{Å}$ cutoff, again separately over the 100 designs generated for each target.

### C.6. Design of D-peptide binders against CD38

We used PDB 7DHA as the reference structure, where chain A is the receptor and chain B and C is the binder that define the binding surface. We sampled 5,000 peptides with the length range of [10,12] under a random seed of 12. These generated complexes were minimized under the Amber ff14sb forcefield, and were filtered and ranked based on the following metrics: **Complementary**. We employed Rosetta to calculate the CavityVolume, ShapeComplemantary, ElectrostaticComplemantary (based on APBS), BuriedHBonds and ExposedHydrophobic, where the latter two are for evaluate the HydrophobicComplemantary. **Interactions**. We employed AutoDock Vina to roughly check the binding energy of each interface, and used PLIP to analyze the interactions between targets and binders. In detail, we counted the total number of identified interactions, the number of hydrogen bonds, and the number of mainchain hydrogen bonds. Finally, we utilized FreeSASA to calculate absolute binding surface area (absBSA), and the ratio of absBSA in the SASA of ligands, termed relative BSA (relBSA). **Conservation**. By the PLIP analysis mentioned above, we identified top10 residues of the receptor that participate the most interactions, as well as the proportions of every interaction type for each residue. These residues are termed "hotspots". Then, for each ligand, we checked how many hotspots it covers, and whether the interaction is the most seen type. For comparison, we reported the weighted coverage, where coverage $C = \sum_i w_i \cdot x_i$, where $x_i$ is a hotspot residue, and the weight $w_i = \frac{N(T_{ij})}{\sum_j N(T_{ij})}$, where $N(T_{ij})$ is the number of interactions of type j for hotspot i.

These metrics were devided into two classes. The first class is for filtering out structures that are not reasonable, where we applied thresholds as follows: absBSA > 400, 0.20 < relBSA < 0.85, vina score < -4.0, BuriedHBonds < 10, ShapeComplemantary > 0.65, ElectrostaticComplemantary > 0.65, TotalInteraction > 8, TotalHBond > 3. In addition, we require CavityVolume and ExposedHydrophobic to fall within the lower 80% of the distribution to exclude interfaces with large voids or excessive exposed hydrophobic patches. The second class is for enriching candidates with high possiblity of binding in the top, where we calculate the Zscore of the number of mainchain hydrogen bonds and the weighted hotspot

coverage, and ranked structrues based on their sum. Finally, the top 6 candidates were subjected to downstream synthesis and analysis.

All peptides are chemically synthesized using the routine Fmoc solid phase peptide synthesis (SPPS) protocol, purified with high-performance liquid chromatography (HPLC), and lyophilized. The target protein CD38 is recombinantly expressed in HEK293F and is purified by affinity chromatography based on Ni-NTA and His-tag.

We then used biolayer interferometry (BLI) to assess binding. Briefly, peptides and target proteins were prepared in BLI buffer (50 mM HEPES, 150 mM NaCl, 0.5% Tween-20, 0.05 mg/mL BSA). Peptides were serially diluted from $200\,\mu$M using a 3-fold scheme to obtain six concentrations, and the target protein was used at $20\,\mu$g/mL. Binding kinetics were quantified by fitting association and dissociation traces to estimate $k_{\mathrm{on}}$ and $k_{\mathrm{off}}$, and $K_D$ was computed as $k_{\mathrm{off}}/k_{\mathrm{on}}$. Besides this kinetic estimation, we also performed steady-state analysis by fitting the equilibrium response at each concentration to a 1:1 binding isotherm,

$$\mathrm{Response} = R_{\mathrm{max}} \cdot \frac{\mathrm{conc}}{K_D + \mathrm{conc}}\,, \tag{63}$$

where $\mathrm{conc}$ denotes the peptide concentration and $R_{\mathrm{max}}$ is the maximal binding response.

For D-1412, the kinetic fitting yields $K_D = 9.9 \pm 0.2\,\mu$M, while the steady-state analysis gives $K_D = 10.5 \pm 0.6\,\mu$M, demonstrating good agreement between the two estimation procedures. For the other 11 candidates, 3 show weak responses but lack enough confidence to claim binding.

