# OpenReview forum: "Cross-Chirality Generalization by Axial Vectors for Hetero-Chiral Protein-Peptide Interaction Design"
_ICML.cc/2026/Conference — ICML 2026 regular_

### Official Review · Reviewer_18oT · 2026-03-11

**Soundness:** 3
**Presentation:** 3
**Significance:** 4
**Originality:** 4
**Overall Recommendation:** 5
**Confidence:** 4

**Summary:**

The paper addresses D-peptide binder design for L-protein targets — a therapeutically important problem (D-peptides resist proteases and have low immunogenicity) that current ML models cannot handle because training data is exclusively homo-chiral (L-L). The key insight is that E(3)-equivariant models are inherently chirality-blind (Proposition 3.2), while frame-based models hard-code L chirality. The authors propose AFI (Axial Feature Injection), which mixes polar vector features with axial vector features (cross products, triple products, commutators derived from SO(3) irreducible decompositions) to break inversion symmetry while preserving SE(3)-equivariance. This is implemented in the UniMoMo latent diffusion framework as PepMirror. Theorems establish that AFI induces chirality-aware representations with stable L/D clustering, and a Wasserstein continuity result guarantees that similar latent codes produce similar generated structures. PepMirror outperforms 10+ baselines in silico and produces a wet-lab validated D-peptide binder against CD38 (KD ~ 10 µM, 1/12 candidates).

**Compliance With Llm Reviewing Policy:**

Affirmed.

**Final Justification:**

PepMirror addresses a genuinely novel problem — D-peptide binder design for L-protein targets — that existing ML models cannot handle because all training data is homo-chiral. The key insight that E(3)-equivariant models are inherently chirality-blind (Proposition 3.2) is simple but consequential, and the fix — injecting axial vector features (cross products, triple products,
  Clebsch-Gordan-derived parity-odd terms) to break inversion symmetry while preserving SE(3)-equivariance — is theoretically elegant and practically lightweight.

  This paper stands out for combining strong theory with wet-lab validation: a D-peptide binder against CD38 at KD ~10 µM from 12 synthesized candidates, with consistent BLI kinetics. Most ML protein design papers stop at in-silico metrics. The latent space analysis (4 orders of magnitude more L/D discrepancy with AFI) empirically validates the theoretical claims, and the finding that Rosetta
   scores are inconsistent between enantiomers is a valuable contribution to the field.

  The rebuttal was thorough and direct. The affinity maturation roadmap (alanine scan + ML-based active learning) is sensible for bridging the 10 µM to therapeutic nM gap. The architecture portability discussion — direct for PaiNN/GMN, indirect for EGNN, redundant for GCPNet — was exactly the level of detail needed. The commitment to clarify that "cross-chirality generalization" means
  handling inverted inputs rather than learning D-L interaction physics addresses the framing concern. The honest acknowledgement that the single-layer theory doesn't extend to the full trained network (W3) is appropriate — this is an open problem, not a flaw.

  The remaining limitations — modest binding affinity, indirect mirror-image design procedure, and the theory-practice gap in the multi-layer setting — are openly discussed and represent natural next steps rather than fundamental weaknesses. This is the strongest paper in my batch: a novel problem, a clean solution grounded in classical physics and representation theory, comprehensive
  evaluation, and wet-lab validation. It merits acceptance.

**Key Questions For Authors:**

1. What is the path from 10 µM to therapeutically relevant affinity? Have you explored affinity maturation starting from D-1412?
2. Could AFI be applied to other equivariant architectures beyond EPT (e.g., GCPNet, EGNN)? Any preliminary results?
3. How does performance scale with peptide length? The CD38 binder is a 10-mer — is there a length regime where AFI's advantages diminish?

**Limitations:**

Yes. Section 5 honestly discusses: (1) the gap between single-layer AFI theory and end-to-end network analysis, and (2) the non-exhaustive search over axial feature constructions. The paper could additionally discuss the modest binding affinity (10 µM vs therapeutic nM) and the indirect nature of the design procedure (mirror-image display rather than direct D-L modeling).

**Strengths And Weaknesses:**

## Strengths

- **Novel and therapeutically important problem with a clean solution.** The observation that E(3)-equivariance implies chirality blindness (Prop. 3.2) is simple but consequential — it explains why existing models fail at hetero-chiral design. Injecting axial vectors to break parity while preserving SE(3)-equivariance is an elegant fix grounded in classical physics (polar vs. axial vectors) and SO(3) representation theory (Clebsch-Gordan decomposition to identify low-order parity-odd features).

- **Wet-lab validation.** A D-peptide binder against CD38 with KD ~ 10 µM from 12 synthesized candidates is a meaningful first-generation result. Most ML protein design papers stop at in silico evaluation. This experimental validation, combined with consistent BLI kinetics and steady-state fitting, substantially strengthens the paper.

- **Thorough evaluation and valuable analysis.** The latent space analysis (Fig. 3-4) empirically validates the theory: AFI produces 20 tight amino acid clusters with L/D enantiomers nearby but distinguishable (4 orders of magnitude more discrepancy than without AFI). The discovery that Rosetta scores are inconsistent between enantiomers (Table S4, Appendix A.4) is an important finding for the field, and the switch to Vina for evaluation is well-justified.

- **Lightweight and general-purpose.** AFI is a plug-and-play module that can be added to any scalarization-based E(3)-equivariant model with minimal overhead. Three different axial feature constructions all work comparably, suggesting robustness to the specific choice.

---

## Weaknesses

- **The binding affinity is modest.** KD ~ 10 µM is far from therapeutic-grade (typically nM). 1/12 hit rate is reasonable for a first campaign but the paper could be more transparent about this gap and the optimization path forward (affinity maturation, longer peptides, etc.).

- **The design procedure is indirect.** PepMirror doesn't directly model D-L interactions — it inverts the L-target, designs an L-peptide for the D-target, then inverts back (mirror-image display). AFI's role is to make the model handle the inverted input correctly, not to model hetero-chiral contacts directly. This is a pragmatic and valid approach, but the framing should be clearer that the generalization is about handling inverted inputs, not about learning D-L interaction physics.

- **Theoretical guarantees are probabilistic with acknowledged gaps.** Theorem 3.1 holds "with probability at least 1 - delta(epsilon)" over random mixing parameters, and Eqs. 6-7 show that specific parameter configurations can eliminate the discrepancy. The paper is honest about this (Section 3.3), but the end-to-end story — from single-layer AFI theory to multi-layer trained model generalization — remains incomplete, as the authors acknowledge in Section 5.

---

> ### Author Rebuttal · Authors · 2026-03-28
>
> ## Response to Weakness 1 and Question 1
> Thank you for the question. Yes, we are currently exploring paths to achieve clinically acceptable affinity in two possible ways.
>
> - **First**, we are trying the conventional affinity maturation process in wet-lab, where we firstly identify hotspots and "coldspots" residues by alanine scan, and construct peptide mutant libraries accordingly. By iterative library screening, higher affinity is expected to be achieved.
>
> - **Second**, we are also exploring an ML-based approach. Current ML methods still struggle to accurately predict peptide–protein binding affinity, and one major reason we attribute to is the lack of reliable, assay-aligned affinity data. We are therefore exploring reinforcement learning and active learning based strategies to acquire new affinity measurements, with the goal of building better affinity prediction models and ultimately enabling in silico affinity maturation.
>
> As for D-1412, we have not tried affinity maturation on this peptide because this is an ability demo on the model target CD38. We are designing peptide binders for more clinically relevant targets, and planning to perform affinity optimization in these campaigns.
>
> ## Response to Question 2
> In principle, AFI is not specific to EPT. It can be extended to other equivariant backbones whenever axial features can be constructed from the original features.
>
> For scalarization-based models that explicitly maintain vector channels, such as PaiNN or GMN style backbones, this extension is direct. One can construct axial features from node vector by cross-channel cross product, and inject them back into the original vector features before scalarization or gating. This is the setting in which AFI is naturally plug-and-play.
>
> AFI can also apply to EGNN though it is less direct, because EGNN does not maintain an explicit node vector feature. Instead, an AFI-style variant would first construct local axial features from multiple relative displacement vectors around each node, and then inject the resulting pseudo-vectors into the message function or coordinate update. In this way, EGNN can in principle be turned into a chirality-aware variant.
>
> GCPNet is already SE(3)-equivariant and chirality-sensitive following a different route. It builds local frames from relative directions and cross products, and repeatedly injects these frame features into scalar and vector updates during message passing. Compared with GCPNet, AFI does not require redesigning the whole feature parameterization. It is intended as a lighter plug-and-play modification to existing backbones.
>
> Nonetheless, we do not yet have any preliminary empirical results on other backbones. We hope AFI can be applied in other backbones for other tasks in future works.
>
> ## Response to Question 3
> We believe the main factor limiting the reliable generation in terms of peptide length is the size distribution of the training data. Since our goal is peptide design, PepMirror is trained on PepBench where peptide length is ranging from 4 to 25, and is evaluated on the LNR test set with lengths from 3 to 27. In this range, we did not observe an obvious trend that generation quality changes with peptide length. The CD38 binder was designed as a 10-mer mainly for practical reasons: shorter peptides are easier to synthesize, and a 10-mer was sufficient to cover the selected epitope.
>
> ## Response to Weakness 2
> Thank you for your advice.
>
> Yes, by claiming *cross-chirality generalization*, we mean the model can generalize from training data conditioned on L-targets to design tasks conditioned on D-targets. We make L and D amino acids have similar but distinguishable representations to enable such transfer. Though the generalization is supported by experiments, we agree that this does not mean the model has learned the shared chemical rules, nor that it directly models D-L interaction physics. We will clarify this point in the final version.
>
> For a more direct design pipeline, a possible way is to globally inverted L-L complex data to D-D complex data and use these inverted structures for training. The model could thus directly generate D-peptides for L-targets.
>
> ## Response to Weakness 3
> While Theorem 3.1 supports the single-layer mechanism, extending it to the trained multi-layer setting is mathematically much more complicated. We hope this gap can be further narrowed in future work. We are also exploring simplified architectures that may preserve good empirical performance while being easier to analyze theoretically.

---

> > ### Author Rebuttal · Reviewer_18oT · 2026-03-31
> >
> > I have read the author response and find it thorough and satisfying.
> >
> >   W1 / Q1 (Affinity path). The two-pronged approach — conventional alanine scan + library screening, and ML-based affinity prediction with active learning — is a sensible roadmap. The honest acknowledgement that D-1412 was a proof-of-concept rather than a clinical candidate is appreciated. No concerns here.
> >
> >   Q2 (Other architectures). The breakdown by architecture type is exactly what I was looking for. The distinction between direct applicability (PaiNN/GMN with explicit vector channels), indirect extension (EGNN via local axial features from displacement vectors), and redundancy (GCPNet already chirality-sensitive via local frames) is clear and useful. The absence of empirical results on other backbones is understandable for this paper's scope.
> >
> >   Q3 (Peptide length). Good to know that no length-dependent quality degradation was observed within the 3–27 range of PepBench/LNR. The 10-mer choice being driven by synthesis practicality rather than model limitation is a helpful clarification.
> >
> >   W2 (Indirect design). The commitment to clarify that "cross-chirality generalization" means transfer from L-target conditioning to D-target conditioning — not learning D-L interaction physics — addresses my concern. The mention of globally inverted L-L → D-D training data as a more direct alternative is an interesting future direction worth noting in the revision.
> >
> >   W3 (Theory gap). Acknowledged. This remains an open problem and the authors are straightforward about it.
> >
> >   Overall: The responses are direct, honest, and add useful detail on all three questions. Nothing in the rebuttal changes my assessment of the paper's strengths or limitations. I maintain my recommendation of 5 (Accept). This remains the strongest paper in my batch — the combination of a novel and well-motivated problem, clean theoretical grounding, comprehensive evaluation, and wet-lab validation is compelling. The limitations (modest affinity, indirect procedure, single-layer theory) are honest about what's next rather than fundamental flaws.

---

> > > ### Author Response · Authors · 2026-04-06
> > >
> > > Thank you again for reviewing our work. We feel encouraged that our response is satisfying. We will revise our manuscript in the final version according to our discussions.

---

### Official Review · Reviewer_6fzA · 2026-03-12

**Soundness:** 3
**Presentation:** 3
**Significance:** 3
**Originality:** 2
**Overall Recommendation:** 4
**Confidence:** 4

**Summary:**

The paper addresses a critical challenge in therapeutic design: the *de novo* design of D-peptide binders for L-protein targets. The authors propose Axial Feature Injection (AFI), a method to transform E(3)-equivariant models into SE(3)-equivariant (chiral-sensitive) ones by injecting axial vectors (e.g., cross-products). They implement this via PepMirror, a latent diffusion model. The work validates the approach through a wet-lab experiment targeting the CD38 protein.

**Compliance With Llm Reviewing Policy:**

Affirmed.

**Final Justification:**

Although the novelty of the paper is limited, primarily incorporating the AFI module, which has been shown to be effective in chirality-related tasks, into the existing UniMoMo model to address the binder design problem, the experimental evaluation is thorough, and the improvements are significant. Moreover, the rebuttal has adequately addressed my concerns. Therefore, I am inclined to raise my score to 4.

**Key Questions For Authors:**

See weaknesses.

**Limitations:**

Yes.

**Strengths And Weaknesses:**

**Strengths:**
1. The paper is well-written and easy to follow.
2. The proposed AFI features are effective in handling peptide chirality.
3. The paper provides wet-lab validation, which strengthens the empirical claims.

**Weaknesses:**
1. While AFI is effective, the concept of using axial vectors to handle chirality is known in the geometric ML community. The novelty lies more in the application to peptide design than in the ML community.
2. Mirror symmetry or central symmetry? The paper's discussion of chirality and symmetry operations appears somewhat unclear. In particular, the distinction between mirror reflection and central inversion is not fully consistent throughout the paper. In Figure 1 (right), the transformation illustrated appears closer to a central inversion (from $X$ to $-X$) than to a mirror reflection across a plane. However, chirality is typically defined with respect to mirror symmetry, not central inversion. This may lead to confusion in interpreting the geometric arguments presented in the paper. Furthermore, the statement in Figure 1, "Axial vectors are invariant under the spatial inversion", is confusing against chiral awareness.
3. Following point 2, the paper's description of whether the backbone model (UniMoMo) is chirality-aware is somewhat confusing. In the contributions, the authors suggest that the E(3)-equivariant backbone is not chiral-sensitive. However, Section 4.2.3 states that inverting the input structure leads to an accordingly inverted output. This behavior suggests that the model is equivariant to inversion rather than completely insensitive to chirality. Furthermore, the paper describes this property as "counterproductive". Then what is the desired property? Is it to keep the output unchanged when the input structure is inverted?
4. Following points 2 and 3, is a chiral-aware property necessary to design a D-peptide? The mirror-image algorithm with a flip [1,2] seems capable of generating D-peptides by applying mirror transformations to L-peptide binders. From Figure 1 (left), it is not entirely clear why such approaches are insufficient in the setting considered here.
5. The necessity of Proposition 3.2 is not entirely clear. The property $c(X)=c(−X)$ appears to follow directly from the symmetry properties of the backbone architecture rather than requiring a separate proposition.
6. The experimental section reports that the model generates approximately 5,000 candidate peptides, from which 12 are selected, and one peptide is good. While such expert filtering is common in practice, this selection process makes it difficult to assess the intrinsic success rate of the generative model itself.

---
[1] Ke Sun, et al. Accurate de novo design of heterochiral protein–protein interactions. *Cell Research* 2024.

[2] Fang Wu, et al. D-Flow: Multi-modality Flow Matching for D-peptide Design. *NeurIPS* 2025 (workshop).

---

> ### Author Rebuttal · Authors · 2026-03-27
>
> ## Response to Weakness 1
> We agree that axial/pseudo-vector concepts themselves are not new in geometric learning. Our claim is that AFI provides a lightweight way to endow E(3)-equivariant backbones with chirality awareness by constructing axial channels from existing vector features.
>
> Prior chirality-aware approaches typically obtain chirality sensitivity by redesigning the representation or message-passing scheme, such as using local frames, geometry-complete features, order-sensitive aggregation, or dedicated chiral embedding blocks. In contrast, AFI modifies the feature interface rather than redesigning the whole backbone, making it plug-and-play for models with vector features. We instantiate this idea on peptide design, where chirality is central, but the mechanism itself is more general and could in principle be extended to other backbones with vector features such as PaiNN- or GMN-style models. We will revise the paper to clarify that our novelty lies in this lightweight injection mechanism, rather than in the general concept of axial vectors itself.
>
> ## Response to Weakness 2
> Sorry for the confusion. Our point is that in $\mathbb{R}^3$,  both mirror reflection and central inversion have a determinant of $-1$ and differ only by a proper rotation, so **they are equivalent for chirality comparison**. We use central inversion in implementation because it flips all three coordinates simultaneously and thus treats them uniformly, whereas a mirror reflection requires choosing a specific mirror plane.
>
> For the statement in Fig. 1, we agree that the phrase “axial vectors are invariant under spatial inversion” can be confusing in isolation. The key point is the parity difference between polar and axial vectors under inversion. Once these two types of channels are mixed, the representation is generically no longer inversion-equivariant, which enables chirality-aware representations.
>
> ## Response to Weakness 3
> By saying UniMoMo is *not chirality-sensitive*, we mean that its residue-level latents do not encode chirality. L- and D-amino acids are mapped to the same latents, as shown theoretically in Eqs. (8) and (9) and empirically in Fig. 3.
>
> You mentioned the observation where inverting the input structure leads to an accordingly inverted output. This is a consequence of E(3)-equivariance at the coordinate level, rather than an evidence of chirality awareness. In other words, the model is sensitive to geometric inversion as a coordinate transformation, but not to chirality as a feature.
>
> The behavior of "ligand chirality co-varies with receptor chirality under inversion" is counterproductive for our task, because our goal is to generate hetero-chiral interfaces rather than preserve homo-chirality after inverting receptor inputs. The desired property is therefore to generate a ligand with the appropriate chirality independently of whether the target is inverted.
>
> ## Response to Weakness 4
> We agree that these prior works are relevant, but they do not remove the central challenge: cross-chirality generalization from predominantly homo-chiral training data.
>
> - **Ke Sun et al.** employs a physics-based method that has a low success rate and **still requires mirror-image screening against D targets**, which relies on total chemical synthesis and thus has limited applications.
> - **Fang Wu et al.** introduced the "flip trick" in ML and provided initial exploration on D-peptide binder design, but no quantitive evaluations are provided for D-peptides. As they discussed in their paper, generated D-peptides “exhibit significantly higher variability” in terms of conformation, and they attributed this to “training distribution mismatch”. Finally, they discussed: “These challenges align with recent findings in protein design that **highlight the challenges of transferring learned molecular representations across different stereochemical spaces**”. In other words, cross-chirality generalization from homo-chiral training data to hetero-chiral design tasks remains unsolved.
>
> In this work, we address this challenge at the residue-representation level, enabling cross-chirality generalization from L-target training data to D-target tasks. This is supported by the latent space analysis and smaller performance drop from L- to D-targets.
>
> ## Response to Weakness 5
> We agree that this proof is straightforward. It was included for completeness in the analysis, and we will reclassify it as a Fact rather than a Proposition.
>
> ## Response to Weakness 6
> We agree that the wet-lab section does not estimate an unbiased end-to-end hit rate of the generative model. Our major intention here is to demonstrate practical feasibility under a realistic generate-and-rank workflow, while the intrinsic model quality is mainly assessed by the dry-lab evaluations. Nonetheless, since most filters are commonly used previously and show weak correlation with affinity, we believe the successful design shows the ability of PepMirror to some extent.

---

> > ### Author Rebuttal · Reviewer_6fzA · 2026-04-02
> >
> > I thank the authors for the detailed rebuttal. My most concerns have been adequately addressed, and I will raise my score accordingly.
> >
> > - Regarding the AFI Module, the clarification that the contribution lies in the lightweight, plug-and-play nature of the injection mechanism rather than in the general concept of axial vectors itself is clear. However, since it is a plug-and-play module, can the authors provide empirical results or further discussion on applying AFI to other suitable backbones? Specifically, where exactly should the module be injected in different architectures? Is the injection method backbone-specific, and does it require extensive hyperparameter tuning? Providing practical guidelines on adapting AFI to other models would strengthen the claim of generalizability.
> >
> > - I understand the explanation that both operations have a determinant of -1 and are thus equivalent for chirality comparison, and that central inversion is chosen for implementation convenience. Instead of restricting the operation to central inversion, what if we apply a randomly sampled transformation with determinant -1 (e.g., a mirror reflection across a randomly chosen plane passing through the molecule's center)? Furthermore, since the current implementation uses central inversion during training, would substituting it with a specific mirror reflection (e.g., mirroring across the xy-plane) during testing/inference affect the performance? It would be valuable to know if this holds empirically. I suggest the authors test this property.

---

> > > ### Author Response · Authors · 2026-04-06
> > >
> > > We are encouraged to hear that we have addressed most of the concerns.
> > >
> > > ### **(1) Extension of AFI to other backbones**
> > >
> > > We believe AFI can be extended to other backbones whenever features with different parity, such as axial vectors or pseudo-scalars, can be constructed from the original features.
> > >
> > > For backbones that explicitly maintain vector channels, such as PaiNN [1] or GMN [2] style models, the extension is straightforward. One can construct axial features from existing vector features as described in our paper, and inject them back into the original vector features before scalarization or gating.
> > >
> > > For backbones that do not explicitly maintain node vector features, such as EGNN [3], the application of AFI is still feasible. For example, one could first construct local axial vectors from displacement vectors around each node, and then inject them into the message function or coordinate update.
> > >
> > > We have tested some relevant AFI variants. In addition to projecting triple scalar products into vector channels, we tested injecting them directly into node scalar features (denoted as triple_scalar). We also tested a lightweight variant that applies such mixing only once after GNN-based feature initialization (denoted as triple_scalar_once). These variants still achieve similarly strong performance, while one-time injection leads to a slight drop in chirality consistency. These results suggest that AFI does not have to rely on vector channels or EPT. Pseudo-scalar features can be easily constructed from a GNN, which can be injected into architectures without vector channels.
> > >
> > > |Operation|Task|Chirality Raw|Chirality Minimized|Vina Suc.|Vina Avg|Vina Top|Vina IMP|
> > > |:-:|:-:|:-:|:-:|:-:|:-:|:-:|:-:|
> > > |triple_scalar|L|99.50|99.37|99.57|-4.31|-5.91|76.34|
> > > |triple_scalar|D|99.33|99.19|99.61|-4.22|-5.88|68.82|
> > > |triple_scalar_once|L|98.04|97.99|99.72|-4.25|-5.88|72.04|
> > > |triple_scalar_once|D|97.05|96.98|99.70|-4.17|-5.76|65.59|
> > >
> > > Nonetheless, we do not yet have other preliminary results for other backbones. We hope AFI can be extended to other architectures in future work, and we will discuss more on this in the final version.
> > >
> > > ### **(2) Performance under reflections across different planes**
> > >
> > > We believe this is theoretically addressed by the proof of SE(3)-equivariance in Appendix B.2. To validate this experimentally, we reflected the structures in the LNR test set across the xy, yz, and xz planes. In addition, we randomly sampled three other planes passing through the structural center and reflected the structures across them (denoted as random_1/2/3).
> > >
> > > We evaluated PepMirror(cross) in terms of chirality and interface energy with these six variants of LNR as targets. The result shows highly similar performance across different reflection planes, supporting that reflections across different planes do not affect the generation quality.
> > >
> > > |Operation|Task|Chirality Raw|Chirality Minimized|Vina Suc.|Vina Avg|Vina Top|Vina IMP|
> > > |:-:|:-:|:-:|:-:|:-:|:-:|:-:|:-:|
> > > |original|L|99.93|99.83|99.67|-4.27|-5.81|69.89|
> > > |inversion|D|99.91|99.81|99.76|-4.15|-5.69|63.44|
> > > |reflection_xy|D|99.90|99.82|99.70|-4.18|-5.72|65.59|
> > > |reflection_yz|D|99.92|99.83|99.62|-4.18|-5.73|69.89|
> > > |reflection_xz|D|99.90|99.82|99.70|-4.19|-5.72|65.59|
> > > |random_1|D|99.90|99.83|99.71|-4.19|-5.72|66.67|
> > > |random_2|D|99.91|99.81|99.59|-4.19|-5.74|65.59|
> > > |random_3|D|99.92|99.83|99.70|-4.19|-5.70|65.59|
> > >
> > > Moreover, we evaluated equivariance at the representation level by comparing residue-level latents of the original, inverted, and reflected LNR pockets, using the same protocol as in Fig. 3. The results show that reflections across different planes still separate L/D amino-acids effectively, while preserving consistent residue latents under rotation.
> > >
> > > |Operation|Chirality|Med. Dist. to original|Med. Dist. to inversion|
> > > |:-:|:-:|:-:|:-:|
> > > |original|L|0|1.5e-2|
> > > |inversion|D|1.5e-2|0|
> > > |reflection_xy|D|1.5e-2|5.3e-7|
> > > |reflection_yz|D|1.5e-2|5.6e-7|
> > > |reflection_xz|D|1.5e-2|3.9e-7|
> > > |random_1|D|1.5e-2|3.0e-5|
> > > |random_2|D|1.5e-2|3.1e-5|
> > > |random_3|D|1.5e-2|3.1e-5|
> > >
> > > We noted that randomly sampled planes show larger deviations from inversion than axis-aligned reflections. We believe this mainly comes from the limited precision of the PDB format, where coordinates are typically stored with three decimal places. Arbitrary-plane reflections are therefore accumulate larger rounding errors, whereas axis-aligned reflections are essentially sign flips and are numerically more stable. Since PDB is commonly used for protein structures, we view this as a realistic setting. Under this precision, AFI preserves rotation equivariance and maintains chirality separation under arbitrary reflections.
> > >
> > > [1] Schütt K. et al., Equivariant message passing for the prediction of tensorial properties and molecular spectra. ICML, 2021.
> > >
> > > [2] Huang W. et al., Equivariant graph mechanics networks with constraints. ICLR, 2022.
> > >
> > > [3] Satorras, V. G. et al., E(n) equivariant graph neural networks. ICML, 2021.

---

### Official Review · Reviewer_TJHf · 2026-03-13

**Soundness:** 3
**Presentation:** 2
**Significance:** 3
**Originality:** 3
**Overall Recommendation:** 4
**Confidence:** 5

**Summary:**

The work present a method to design D-amino-acid peptides, not through mirroring but through combinational signal of axial and polar vectors.

**Compliance With Llm Reviewing Policy:**

Affirmed.

**Final Justification:**

The authors answered my concerns and questions and addressed the limitations. I raise my confidence for the score.

**Key Questions For Authors:**

1. AFI mixes polar and axial channels through linear mixing coefficients. Would other mixing mechanisms (nonlinear, cosine-similarity-based) preserve the theoretical guarantees in Appendix B?
2. Equation (14), what about $d(X,X')$ and $d(X,-X')$? In this case, a D-form residue could be more similar to a different L-form amino acid. In both supporting figures, there appear to be some examples. For the Tanimoto similarity in Figure 2, g and A are more similar. In the latent distance heatmap in Figure 4, g and C are more similar.
3. In Figure 2, amino acid pairs like T and V, Y and F, seem to be more similar. However, the similarity is not reflected in the Figure 4's t-SNE visualization. Could the authors clarify how the latent embedding captures amino-acid similarity?
4. Figure 3 shows results for different AFI constructions among cross, triple, and commutator. Can they be combined to provide complementary information?
5. How are the samples for the t-SNE visualization generated? Are these embeddings of generated peptides, or residue-level embeddings extracted from structures? In Figure S2, many L and D embeddings appear to overlap. Does AFI explicitly separate chirality or primarily encode residue identity?
6. Appendix B.1 when the author defines $\mathbf{f_3}$,
\begin{equation}
 \left[ M(v), M(u) \right] = M(v)M(u)-M(u)M(v) = (u\cdot v)(vu^\top - uv^\top)
\end{equation}
this appears to produce $-(u\cdot v)(u\times v)$ under the ax map. Could the authors clarify if this is a typo or a sign convention?

**Limitations:**

- What about L-D heterochiral designs? This may also be relevant in practice, as D amino acids can be more expensive to synthesize.
- For BLI results, what are the corresponding results for the L-form versions of these sequences? These could serve as useful negative controls.

The core contribution is clear, though several aspects of the analysis and presentation would benefit from clarification.

**Strengths And Weaknesses:**

**Soundness**
Both in silico evaluation and in vitro validation are technically sound.

**Presentation**
 - "Submission and Formatting Instructions for ICML 2026" change your headers into your actual title.
 - In Theroem 3.1 in the main text, $\delta$ and $c$ appear but are not clearly defined. Is this $c$ the same as Equation (3).
 - Figure 5, what is the red part of the protein structure?

**Significance** Important for D-amino-acid peptide design, which has therapeutic potential regarding protease resistance

**Originality**
Injecting axial vector signals into polar-vector E(3)-equivariant architectures to enable chirality sensitivity is a novel idea.

**Strength** Detailed mathematical proofs and experimental design.
Have in vitro experiment to show the actual application.

**Weakness** See questions

---

> ### Author Rebuttal · Authors · 2026-03-28
>
> ## Response to Presentation Issues
> Thank you for the comments. We will replace the headers and clarify that red/blue denotes positive/negative electrostatic surface potential in Fig. 5.
>
> $c$ is the same as defined in Eq. (3), $\delta_W>0$ is a small parameter controlling the high-probability event. Since its full definitions depend on several auxiliary quantities, we refer the reader to the formal statement in Theorem B.8 and Corollary B.9 in the main text. We will make this clearer in the next revision.
>
> ## Response to Question 1
> Our Appendix B guarantees rely on different properties to different extents. For SE(3)-equivariance and parity consistency, linearity is sufficient but not necessary: other mixers could also preserve these properties if they are symmetry-compatible. However, the main chirality-awareness guarantee in Appendix B.3 is more specific to the linear mixer. The proof uses the explicit decomposition under inversion. For generic nonlinear mixing, this explicit bilinear reduction is generally unavailable, so the same theorem does not follow directly. We use linear mixing since it is a minimal, interpretable mechanism that isolates the AFI effect and admits clean proof. We will try extending the guarantees to more complex mixers in future work.
>
> ## Response to Question 2
> First, we would like to clarify that:
> - Because Glycine is achiral, g and G are actually identical. This is why we masked $(g,G)$ and $(G,g)$ entries along with diagonal entries in Fig. 2.
> - For the heatmap in Fig. 4, the $(g,C)$ entry actually corresponds to a large distance, indicating that their latents are *not* similar.
>
> That said, the Tanimoto similarity does show some ambiguity. Therefore, we use it just as a coarse motivation, rather than direct evidence for Eq. (14). The numerical support for Eq. (14) is provided in the latent distance heatmap in Fig. 4, where it is clear that $d(X,–X')>d(X,–X)$.
>
> ## Response to Question 3
> **First**, t-SNE does not preserve distances, so the left panel of Fig.4 is a *qualitative* visualization. A *quantitative* comparison is provided as a heatmap in the right panel of Fig.4.
>
> **Second**, the final latent is contextualized by the environment and training objective, so Tanimoto similarity cannot determine latent distances and only serves as an indirect corroboration of Eq. (14). We will revise contents to avoid overstating the significance of Fig. 2.
>
> ## Response to Question 4
> We tested a combined version that mixed cross, triple, and commutator features with the original polar vector, and it did not show a clear improvement on LNR. However, this is still an interesting direction, and we will test other ways for axial feature construction.
> |Task|raw chirality|minimized chirality|Suc|Avg.|Top|IMP|
> |:-:|:-:|:-:|:-:|:-:|:-:|:-:|
> |L|99.95|99.86|99.61|-4.27|-5.89|72.83|
> |D|99.97|99.87|99.70|-4.18|-5.75|75.00|
>
> ## Response to Question 5
> The t-SNE samples are residue-level embeddings extracted from pockets in LNR, not generated peptides. Fig. S2 is only meant to show that AFI prevents L/D residues from collapsing to identical latents. In practice, AFI adds chirality information while the reconstruction objective preserves residue identity, the combined effect is that the model becomes chirality-aware while still maintaining stable latents for each amino-acid type, as supported by Fig. 4.
>
> ## Response to Question 6
> This is a sign-convention issue. In Appendix B.1 we define $ax(A)=a$ via $A_{ij}=\epsilon_{ijk}a_k$, which implies $Ax=-a\times x$. Hence, from$[M(v),M(u)]x=(u\cdot v)(u\times v)\times x$ one gets$\operatorname{ax}([M(v),M(u)])=-(u\cdot v)(u\times v)$. So the reviewer is correct: Eq. (31) should carry a minus sign. We will clarify this in the revision. Importantly, this does not affect the later theory, since only the axial equivariant nature of $f_3$ is used, and a global sign can be absorbed by the subsequent linear mixing.
>
> ## Response to Limitation 1
> Partial D-substitution may be useful in practice, but choosing how many residues to invert and where to place them remains highly case-dependent. We focus on all-D peptides because their guaranteed stability allows focusing on optimizing other properties.
>
> ## Response to Limitation 2
> Thank you for the advice. We agree the L-form results are valuable. However, recent evidence shows that enantiomers can both retain binding, and the affinity gap decreases as the structure gets more disordered[1]. L-enantiomers are therefore not determined to be negative controls. After testing the L-form of D-1412, we observed binding as well ($K_D\approx15\ \mu$M). Considering D-1412 is a short peptide without secondary structures, such behavior is plausible.
>
> Moreover, we believe the L-form results do not affect our claim: PepMirror generates plausible D-peptide binders for L-targets, as supported by binding energy evaluations. Whether L-enantiomers also bind to targets does not undermine the validity of D-binders.
>
> [1] Nature, 2024, 636, 762–768.

---

> > ### Author Rebuttal · Reviewer_TJHf · 2026-04-03
> >
> > Thank the authors for the clarification.

---

> > > ### Author Response · Authors · 2026-04-06
> > >
> > > Thank you again for reviewing our work, and we are encouraged that concerns have been addressed.

---

### Official Review · Reviewer_C2Eu · 2026-03-13

**Soundness:** 3
**Presentation:** 2
**Significance:** 3
**Originality:** 3
**Overall Recommendation:** 4
**Confidence:** 4

**Summary:**

The manuscript presents PepMirror, a latent diffusion framework aimed at the de novo design of D-peptide binders for L-protein targets, a task complicated by the "chiral gap" in available structural data. The core technical contribution is the Axial Feature Injection (AFI) module, which aims to break the mirror symmetry of E(3)-equivariant models to achieve SE(3) equivariance (chirality awareness). The authors validate their method using in silico metrics and provide experimental evidence through the design of CD38-binding peptides.

**Compliance With Llm Reviewing Policy:**

Affirmed.

**Final Justification:**

The response addresses my primary concerns, while I suggest some minor refinements for the final version.

**Key Questions For Authors:**

1.  In Figure 1, you describe chirality using central inversion ($X$ to $-X$). However, chirality is formally defined by mirror reflection. Can you clarify this distinction and correct it if needed?
2.  The equivariance proofs in Appendix B.2 focus on translation and rotation but do not cover reflection. Since reflection is key to chirality, can you provide a complete proof that includes reflection?
3.  In Section 4.2.3, you say the E(3) backbone’s behavior is "counterproductive." What would be the ideal behavior for a chiral-aware model in this case?
4.  Regarding the "original UniMoMo" baseline: Was it retrained under the same conditions (e.g., dataset, data augmentation) as PepMirror? Please clarify to ensure a fair comparison.

**Limitations:**

yes

**Strengths And Weaknesses:**

Strengths
1. The inclusion of wet-lab experimental data on the CD38 target elevates the paper's impact.
2. The authors identify the bottleneck of lacking D-L interaction data and propose a principled geometric solution.

Weaknesses
1. The manuscript exhibits a lack of rigor in distinguishing between mirror reflection and central inversion. In Figure 1 (right), the transformation represents a point inversion, whereas chirality is fundamentally defined by the lack of superimposability upon mirror reflection.
2. The equivariance proofs in Appendix B.2 appear incomplete. The derivations primarily focus on translation and rotation, neglecting formal treatment of reflections which are central to the paper's claims. Furthermore, the proof seems to state central symmetry ($X$ to $-X$), which does not fully capture the geometric requirements of chirality.
3. While the application to D-peptides is appealing, the axial vectors for chiral awareness and the "mirror-flip" strategy are well-established concepts in geometric deep learning and computational chemistry, respectively. The contribution is more of an incremental integration for the application than a fundamental algorithmic method.
4. In Section 4.2.3, the authors state that the E(3) backbone's behavior (producing an inverted output for an inverted input) is "counterproductive," yet it is unclear what the "ideal" behavior should be for a truly chiral-aware model in this context. Additionally, the description of the "original UniMoMo" baseline is vague. It is unclear whether this baseline is retrained using the same peptide-only dataset and the same flipping augmentation as PepMirror, or if it is a pre-trained checkpoint. A fair ablation requires keeping all data processing identical except for the AFI module.

---

> ### Author Rebuttal · Authors · 2026-03-27
>
> ## Overall Response
> Thank you for reviewing our work. As you summarized, to achieve hetero-chiral peptide binder design when few hetero-chiral complex data are available, we present AFI for two purposes.
>
> - **First**, AFI breaks the mirror symmetry of E(3)-equivariant models, so that L and D amino acids will have different latents. This makes models trained sorely on L-L data keep generating L binders for D targets, instead of D binders for D targets, which is still homo-chiral and of low-value.
> - **Second**, AFI remains the *stability* of latent embeddings where L and D forms of the same amino acid still cluster together, so that it is possible for the model to generalize from L-targets training data to D-targets tasks.
>
> We notice that there might be some misunderstandings, and we would like to clarify:
> - **Mirror reflection and central inversion are equivalent for chirality comparison**, because the produced structures can be superimposed by rotation, which does not change chirality.
> - **The purpose of AFI is to change E(3)-equivariance to SE(3)-equivariance**, which requires to break inversion/reflection equivariance. As this prevents the model from generating peptides with a chirality co-vaires with that of the receptor.
>
> We will try to make these points more clear in final version, and please refer to responses below for detailed discussions.
>
> ## Response to Weakness 1 & Question 1
> Thank you for pointing this out. In $\mathbb{R}^3$, mirror reflection and central inversion are distinct operations, but both have a determinant of $-1$ and differ only by a proper rotation, so **they are equivalent for chirality comparison**. In our implementation we use central inversion because it flips all three coordinates simultaneously and thus treats them uniformly, whereas a mirror reflection requires choosing a specific mirror plane. We will clarify this in the future version. We would like to mention that using central inversion to change chirality is also used by other researchers. [1]
>
> ## Response to Weakness 2 & Question 2
> As we discussed in Section 3.3., the purpose of AFI is to break the inversion equivariance of vector features and keep the SE(3)-equivariance at the same time, so that the model can be aware of the difference of chirality in residue-level representations. Since reflection and inversion produce equivalent structures, **our purpose is also to break the reflection equivariance**. Therefore, equivariance proofs in Appendix B.2 should be regarded as complete in this context. Our Proposition 3.5 claims the SE(3)-equivariance, which contains only translation and rotation.
>
> ## Response to Weakness 4 & Question 3
> The behavior of "ligand chirality co-varies with receptor chirality under inversion" is counterproductive for our task, because our goal is to generate hetero-chiral interfaces rather than preserve homo-chirality after inverting receptor inputs. The desired property is therefore to generate a ligand with the appropriate chirality independently of whether the target is inverted. AFI is introduced precisely for this purpose: it resolves the chirality ambiguity of the vanilla E(3) representation while preserving SE(3)-equivariance, thereby enabling chirality-aware modeling and consistent chirality in generated peptide structures.
>
> ## Response to Weakness 4 & Question 4
> We provided two baselines related to UniMoMo in the submitted paper.
> - The UniMoMo(all) is the published version that has not been retrained, so its training data include small molecules, peptides, and nanobodies.
> - The UniMoMo(pep.) is the retrained version under the same conditions as PepMirror for a fair comparison.
>
> ## Response to Weakness 3
> We agree that axial/pseudo-vector concepts themselves are not new in geometric learning. Our claim is that AFI provides a lightweight way to endow E(3)-equivariant backbones with chirality awareness by constructing axial channels from existing vector latents.
>
> Prior chirality-aware approaches typically obtain chirality sensitivity by redesigning the representation or message-passing scheme, such as using local frames, geometry-complete features, order-sensitive aggregation, or dedicated chiral embedding blocks. In contrast, AFI modifies the feature interface rather than redesigning the whole backbone, making it plug-and-play for models with vector features. We instantiate this idea on peptide design, where chirality is central, but the mechanism itself is more general and could in principle be extended to other backbones with vector features such as PaiNN- or GMN-style models. We will revise the paper to clarify that our novelty lies in this lightweight injection mechanism, rather than in the general concept of axial vectors itself.
>
> [1] Ke Sun, et al. Accurate de novo design of heterochiral protein–protein interactions. Cell Res. 2024.

---

> > ### Author Rebuttal · Reviewer_C2Eu · 2026-04-03
> >
> > I thank the authors for the detailed response. The clarification addresses my primary concerns. I will raise my score accordingly and suggest the following minor refinements for the final version:
> >
> > 1. **Robustness to General Reflections:** While central inversion and mirror reflection are equivalent for chirality, could the authors clarify if the model remains robust under **arbitrary mirror planes** (any $O\in O(3)$ with $det=-1$) during testing? Ensuring performance isn't tied to the specific inversion center would strengthen the "chiral awareness" claim.
> > 2. **Theoretical Completeness:** Regarding Appendix B.2, I agree that proving **SE(3)-equivariance** (rotation/translation) is sufficient for the task. However, explicitly formalizing how axial vectors transform under reflection to "break" E(3) symmetry would make the theoretical framework more rigorous.

---

> > > ### Author Response · Authors · 2026-04-06
> > >
> > > Thank you for reviewing our work, and we are encouraged that primary concerns have been addressed.
> > >
> > > ### **(1) Robustness to General Reflections**
> > >
> > > We believe this is theoretically addressed by the proof of SE(3)-equivariance in Appendix B.2. To validate this experimentally, we reflected the structures in the LNR test set across the xy, yz, and xz planes. In addition, we randomly sampled three other planes passing through the structural center and reflected the structures across them (denoted as random_1/2/3).
> > >
> > > We evaluated PepMirror(cross) in terms of chirality and interface energy with these six variants of LNR as targets. The result shows highly similar performance across different reflection planes, supporting that reflections across different planes do not affect the generation quality.
> > >
> > > |Operation|Task|Chirality Raw|Chirality Minimized|Vina Suc.|Vina Avg|Vina Top|Vina IMP|
> > > |:-:|:-:|:-:|:-:|:-:|:-:|:-:|:-:|
> > > |original|L|99.93|99.83|99.67|-4.27|-5.81|69.89|
> > > |inversion|D|99.91|99.81|99.76|-4.15|-5.69|63.44|
> > > |reflection_xy|D|99.90|99.82|99.70|-4.18|-5.72|65.59|
> > > |reflection_yz|D|99.92|99.83|99.62|-4.18|-5.73|69.89|
> > > |reflection_xz|D|99.90|99.82|99.70|-4.19|-5.72|65.59|
> > > |random_1|D|99.90|99.83|99.71|-4.19|-5.72|66.67|
> > > |random_2|D|99.91|99.81|99.59|-4.19|-5.74|65.59|
> > > |random_3|D|99.92|99.83|99.70|-4.19|-5.70|65.59|
> > >
> > > Moreover, we evaluated equivariance at the representation level by comparing residue-level latents of the original, inverted, and reflected LNR pockets, using the same protocol as in Fig. 3. The results show that reflections across different planes still separate L/D amino-acids effectively, while preserving consistent residue latents under rotation.
> > >
> > > |Operation|Chirality|Med. Dist. to original|Med. Dist. to inversion|
> > > |:-:|:-:|:-:|:-:|
> > > |original|L|0|1.5e-2|
> > > |inversion|D|1.5e-2|0|
> > > |reflection_xy|D|1.5e-2|5.3e-7|
> > > |reflection_yz|D|1.5e-2|5.6e-7|
> > > |reflection_xz|D|1.5e-2|3.9e-7|
> > > |random_1|D|1.5e-2|3.0e-5|
> > > |random_2|D|1.5e-2|3.1e-5|
> > > |random_3|D|1.5e-2|3.1e-5|
> > >
> > > We noted that randomly sampled planes show larger deviations from inversion than axis-aligned reflections. We believe this mainly comes from the limited precision of the PDB format, where coordinates are typically stored with three decimal places. Arbitrary-plane reflections are therefore accumulate larger rounding errors, whereas axis-aligned reflections are essentially sign flips and are numerically more stable. Since PDB is commonly used for protein structures, we view this as a realistic setting. Under this precision, AFI preserves rotation equivariance and maintains chirality separation under arbitrary reflections.
> > >
> > > ### **(2) Theoretical Completeness**
> > >
> > > We agree that making the reflection case explicit would improve the rigor. In fact, the reflection version reduces directly to our inversion-based analysis: for $F \in O(3)\setminus SO(3)$ (an orthogonal transform with determinant $-1$, including reflection), let $P=-I$ and define $Q := FP \in SO(3)$. Then $F = QP$, so by SE(3)-equivariance under $Q$, together with the inversion parity rule already used in Appendix B.3, we obtain
> > >
> > > $
> > > v(FX)=v(QPX)=Qv(PX)=-Qv(X),\qquad
> > > a(FX)=a(QPX)=Qa(PX)=Qa(X).
> > > $
> > >
> > > Thus, polar channels transform with a sign flip under reflection, while axial channels do not, exactly as required to break full E(3) symmetry.
> > >
> > > Consequently, for the mixed feature $\tilde v = A^\top v + B^\top a$, if we write $p = A^\top v(X)$ and $q = B^\top a(X)$, then
> > >
> > > $
> > > \tilde v(FX)=Q(-p+q),
> > > \qquad
> > > \|\tilde v(FX)\|=\|-p+q\|,
> > > $
> > >
> > > since $Q \in SO(3)$ preserves norms. Therefore, the discrepancy under reflection is identical to the discrepancy under inversion, and Theorem B.8 extends verbatim to reflections by this reduction. We will add this explicit reduction in the revision to make the theoretical framework fully clear.

---

### Decision · Program_Chairs · 2026-04-30

**Decision:**

Accept (regular)

**Comment:**

This paper presents technically solid paper for cross-chirality generation, which is important for peptide design. All reviewers unanimously vote for acceptance.